# Modelling Torsade de Pointes arrhythmias in vitro in 3D human iPS cell-engineered heart tissue

Masahide Kawatou[1,2], Hidetoshi Masumoto[1,2], Hiroyuki Fukushima[1], Gaku Morinaga[1,3], Ryuzo Sakata[2], Takashi Ashihara[4] & Jun K. Yamashita[1]

Torsade de Pointes (TdP) is a lethal arrhythmia that is often drug-induced, thus there is an urgent need for development of models to test or predict the drug sensitivity of human cardiac tissue. Here, we present an in vitro TdP model using 3D cardiac tissue sheets (CTSs) that contain a mixture of human induced pluripotent stem cell (hiPSC)-derived cardiomyocytes and non-myocytes. We simultaneously monitor the extracellular field potential (EFP) and the contractile movement of the CTSs. Upon treatment with IKr channel blockers, CTSs exhibit tachyarrhythmias with characteristics of TdP, including both a typical polymorphic EFP and meandering spiral wave re-entry. The TdP-like waveform is predominantly observed in CTSs with the cell mixture, indicating that cellular heterogeneity and the multi-layered 3D structure are both essential factors for reproducing TdP-like arrhythmias in vitro. This 3D model could provide the mechanistic detail underlying TdP generation and means for drug discovery and safety tests.

[1] Department of Cell Growth and Differentiation, Center for iPS Cell Research and Application, Kyoto University, 53 Shogoin Kawahara-cho, Sakyo-ku, Kyoto 606-8507, Japan. [2] Department of Cardiovascular Surgery, Kyoto University Graduate School of Medicine, 54 Shogoin Kawahara-cho, Sakyo-ku, Kyoto 606-8507, Japan. [3] Nippon Boehringer Ingelheim Co., Ltd. Kobe Pharma Research Institute, 6-7-5 Minatojima-minamimachi, Chuo-ku, Kobe, Hyogo 650-0047, Japan. [4] Department of Cardiovascular Medicine, Shiga University of Medical Science, Seta Tsukinowa-cho, Otsu, Shiga 520-2192, Japan. Correspondence and requests for materials should be addressed to J.K.Y. (email: juny@cira.kyoto-u.ac.jp)

Cardiac toxicity is the most crucial adverse event in drug discovery and development[1–3]. In particular, drug-induced arrhythmia is one of the most common causes of drug withdrawal from the market[4, 5]. Torsade de Pointes (TdP), a representative drug-induced lethal arrhythmia, is a polymorphic ventricular tachycardia (VT) that is characterized by a twisting wave appearance in electrocardiograms (ECGs) and leads to ventricular fibrillation and sudden death[6]. The ICH S7B guidelines[7], which are currently used for the non-clinical pharmacological safety testing of human pharmaceuticals and include information concerning integrated risk assessments, set QT interval prolongation in ECGs as a major endpoint. This prolongation reflects the delayed ventricular repolarization and is a cause of subsequent TdP.

In addition to in vivo animal tests using canine or monkey under telemetry, the guidelines advocate using mammalian cell lines that constitutively overexpress the human ether-a-go-go related gene (hERG), which encodes the cardiac delayed-rectifying K$^+$ channel (IKr) (hERG test)[7, 8]. Human induced pluripotent stem cell (hiPSC)-derived cardiomyocytes have created the possibility of using human cells to test the arrhythmogenicity of drugs[9, 10]. However, single cell types (cardiomyocytes alone) in two-dimensional (2D) culture-based methods only display restricted abnormal electrical activities, such as the prolongation of field potential duration (FPD) corresponding to the QT interval in an ECG, and transient phenomena such as early after depolarization and triggered activity[11, 12]. Additionally, 2D culture methods fail to show the actual electrical activities of TdP, which include sustained irregular electrical activity due to re-entry of electrical excitation among neighboring cardiac cells. More importantly, these methods fail to reproduce the abnormal kinetics of TdP that occur in native three-dimensional (3D) heart tissue. An in vitro 3D model with human cells that can reproduce TdP has never been reported as far as we know. We hypothesized that reproducing TdP in vitro might be possible if 3D heart tissue could be generated from hiPSCs.

In the present study, we integrate our two unique technologies to systematically induce various cardiovascular cells from hiPSCs[13, 14] and to generate 3D tissue-like structures using a bioengineered cell sheet technology[14–17]. Using these techniques, we generate an in vitro drug-induced TdP model that recapitulates the actual kinetics of TdP only with hiPSC-derived cell populations.

## Results

**Generation of 3D hiPSC-derived cardiac tissue sheets**. First, we tried to generate a 3D model with pure cardiomyocytes. Based on our reported method[13, 14], we prepared pure cardiomyocytes from hiPSCs (836B3 line[18]). In brief, we differentiated hiPSCs toward mesodermal cell lineages using defined chemicals and growth factors in a high-density 2D culture. We purified mesodermal cells (platelet-derived growth factor receptor type alpha-positive) and then further differentiated the mesoderm cells into cardiomyocytes. Highly pure cardiac troponin T-positive cardiomyocytes (96.3 ± 2.5%; flow cytometry) were successfully obtained (Supplementary Fig. 1a–d). The induced cardiomyocytes were mostly ventricular cardiac muscle form of myosin light chain 2 (MLC2V)-positive ventricular-type cardiomyocytes [97.3 ± 1.3% (n = 4)] (Supplementary Fig. 1e, f). Then, we generated a 3D cell sheet structure purely from the cardiomyocytes. We re-cultured the cardiomyocytes on temperature-responsive culture dishes (Supplementary Fig. 1g). After 4 days of culture at 37 °C, we reduced the culture temperature to room temperature and collected the cells as self-pulsating cell sheets exclusively consisting of cardiomyocytes (94.7 ± 3.6% of total cells; flow cytometry). We placed the pure cardiomyocyte sheets

on multi-electrode array probes to measure the EFP after exposure to E-4031, which is a representative IKr channel blocker that typically exhibits cardiotoxicity by delayed ventricular repolarization and QT prolongation on ECG, marking eventual TdP. Even though we observed dose-dependent FPD prolongation, TdP-like waveforms were never induced in the pure cardiomyocyte sheets, even with high doses of E-4031 up to 2 μM, which specifically block IKr channels (Supplementary Fig. 2)[19, 20].

Considering the pathological findings of cardiomyopathy patients with serious ventricular arrhythmia, the heart is deformed and consists of fewer cardiomyocytes and more fibrotic tissue compared with healthy individuals[21–23]. Thus, we hypothesized that increasing the cellular heterogeneity within the heart tissue constructs may better reproduce the arrhythmogenic microenvironment. We planned to reduce the cardiomyocyte component within the cell sheets by including hiPSC-derived non-myocytes. We induced CD90-positive (98.9 ± 0.2%; flow cytometry) mesenchymal cells (also positive for vimentin, α-smooth muscle actin, and calponin via immunostaining) (Supplementary Fig. 3a–g). Mesenchymal cells represent a non-myocyte population within the heart, which include smooth muscle cells and cardiac fibroblasts. We generated cell sheets in which cardiomyocytes and non-myocytes were evenly mixed to prepare CTSs with heterogeneous cellular compositions (Supplementary Fig. 3h). The CTSs consisted of cardiomyocytes (52.7 ± 10.0% of total cells) and mesenchymal cells (39.9 ± 9.0%) according to flow cytometry (Supplementary Fig. 3i). Confocal immunostaining images of CTSs indicated a heterogeneous cellular composition and allocation (Fig. 1a, b). Cross-sectional images revealed that the CTSs had a multi-layered 3D cellular structure of 5–6 cell layers, verifying that the CTSs were constructed as 3D structures (Fig. 1c, d and Supplementary Movie 1). Even after forming 3D structures, the cell viability within the CTSs was extremely high (Supplementary Fig. 4), and no sign of hypoxia was detected based on HIF1α expression (Supplementary Fig. 5). The expression levels of several ion channel genes in the CTSs were comparable with those in adult human heart, except *KCNJ2* (coding Kir2.1; related to IK1 current) and *SCN5A* (coding Na$_V$1.5; related to INa current), suggesting that the CTSs were not fully matured adult cardiac tissue (Supplementary Fig. 6)[24, 25]. The EFP of the CTSs measured on a multi-electrode array indicated comparable electrical waveforms at all electrodes (Fig. 1e, f). An electrical propagation map showed a synchronized, unidirectional propagation pattern (Fig. 1g). Thus, we generated 3D heterogeneous heart tissues entirely from hiPSCs that act as an in vitro beating cardiac wall model.

**Drug-induced TdP-like EFP waveforms in CTSs**. Next, we investigated the occurrence of drug-induced TdP using our in vitro model. We recorded the EFP of CTSs and examined the response toward E-4031. E-4031 treatment (up to 100 nM) led to dose-dependent prolongation of the FPD (Fig. 2a–c), indicating that our CTSs respond to E-4031 in a manner that is comparable to hiPSC-derived single cardiomyocytes[11] and pure cardiomyocyte sheets (Supplementary Fig. 2). The beat rate; inter-spike intervals, which represents the R-R interval in ECGs; conduction velocity; and the first peak amplitude were all unaffected by the E-4031 concentration (Table 1). In addition to FPD prolongation, an irregular waveform was observed at higher doses of E-4031 (higher than 100 nM) (Fig. 2d). We finally observed a sustained waveform of tachyarrhythmia (Fig. 2e) and a characteristic TdP-like pattern with polymorphic twisting waveform (Fig. 2f and Supplementary Movie 2). These results strongly suggest that TdP-

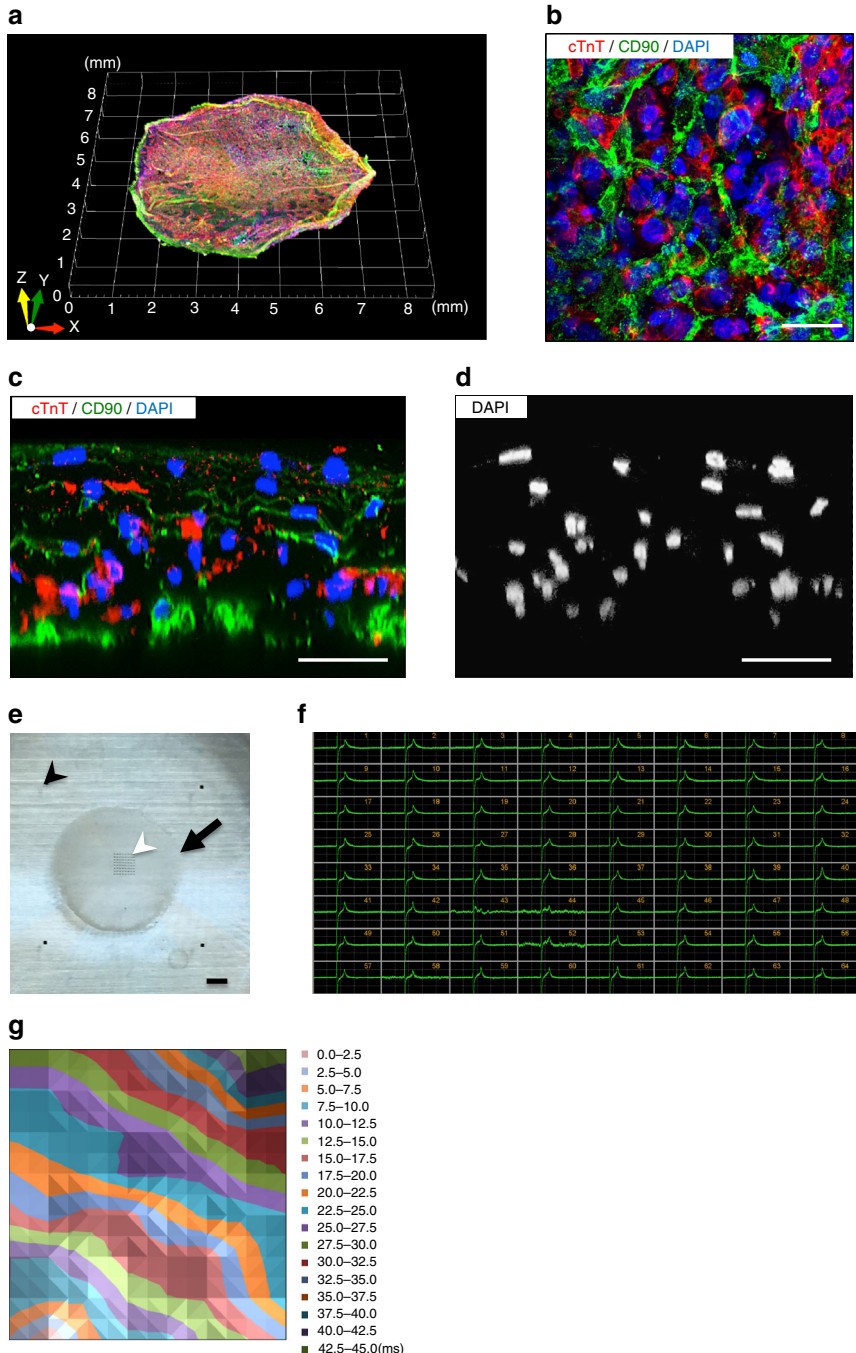

**Fig. 1** Characterization of heterogeneous 3D cardiac tissue sheets (CTSs). **a** Representative confocal microscopic image of a whole CTS that was immunostained for cTnT (cardiomyocytes, red), CD90 (mesenchymal cells, green), and DAPI (nuclei, blue). **b** High-magnification image of a horizontal cross-section (see Supplementary Movie 1). cTnT (red), CD90 (green), and DAPI (blue). **c** Confocal immunostaining of a vertical cross-section. cTnT (red), CD90 (green), and DAPI (blue). **d** Confocal immunostaining of a vertical cross-section. DAPI (white). **e** Macroscopic appearance of a CTS set on a multi-electrode array probe. Arrow: CTS. Black arrowhead: a reference electrode (four reference electrodes in total). White arrowhead: main electrodes (multi-electrode array). **f** Representative EFP waveforms obtained from each electrode. **g** Representative electrical propagation map. Propagation from the lower left corner towards the upper right direction. cTnT, cardiac troponin-T; DAPI, 4′,6-diamidino-2-phenylindole. Scale bars: 25 μm in **b**, 50 μm in **c**, **d**, and 1 mm in **e**

like arrhythmias can be induced in our 3D CTS model. We further tried to confirm the actual cell behavior as tachyarrhythmia.

**Visualization of spiral wave re-entry in TdP model.** Spiral wave re-entry is a considerable mechanism of lethal tachyarrhythmia (Fig. 3a)[26–29]. This theory assumes two representative patterns of

the spiral wave re-entry in tachyarrhythmia: an anchoring (stationary) pattern of the spiral wave center, which results in sustained monomorphic VT, and a meandering (chaotically moving around) pattern of the spiral wave center, which results in polymorphic VT such as TdP. To visualize and confirm the occurrence of TdP-like arrhythmias as 2D spiral waveforms of the actual cell kinetics, we employed a high-precision live cell motion

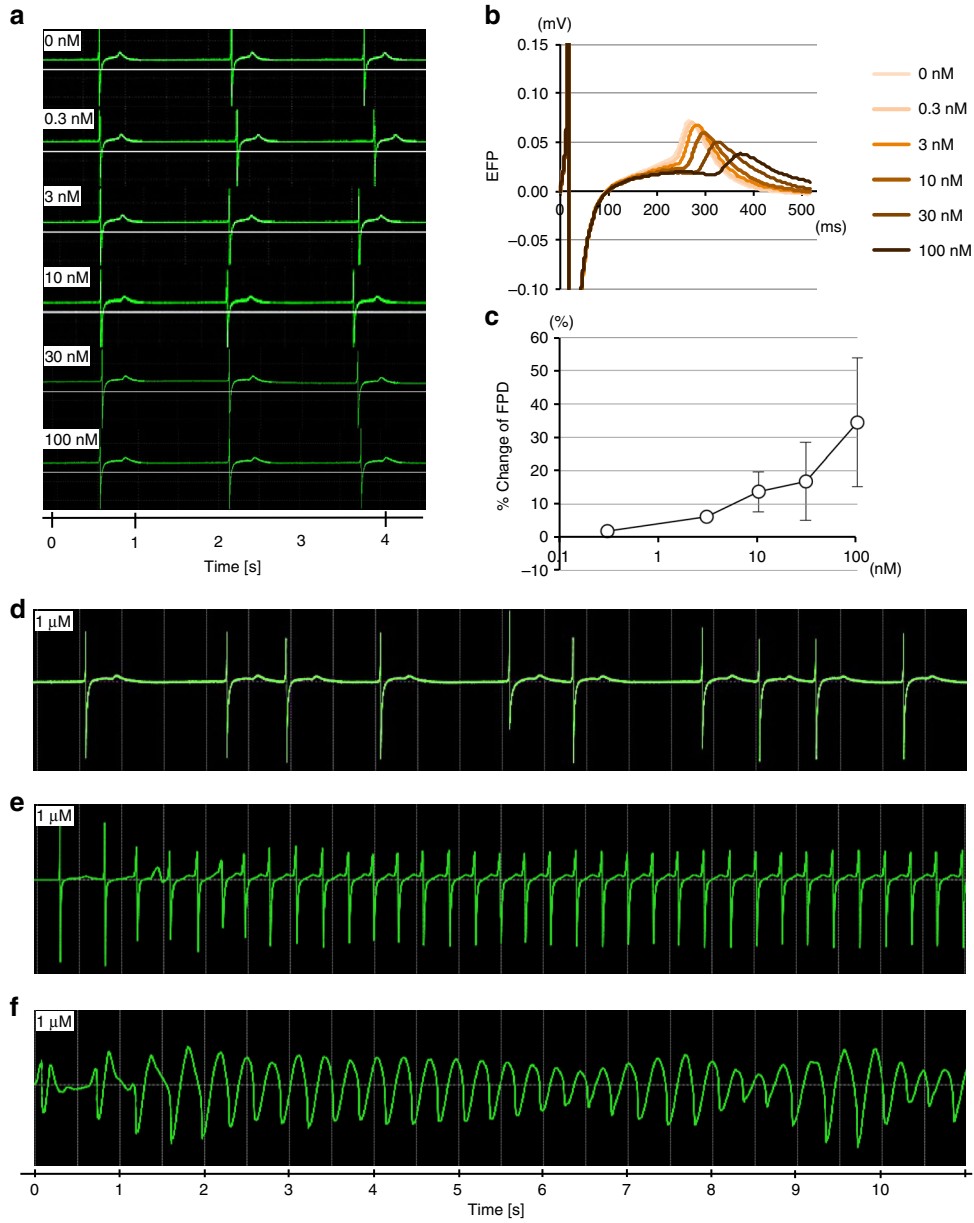

**Fig. 2** Drug-induced TdP-like EFP waveforms in CTSs. **a** EFP waveforms in response to E-4031 (IKr channel blocker). **b** Overlaid EFP waveforms at each E-4031 dosage. FPD prolongation is observed. **c** Relative change in FPD after E-4031 treatment (±s.d., n = 4). **d** Representative transient change of EFP waveforms at an early stage after E-4031 treatment (1 μM). **e** Representative monomorphic ventricular tachycardia-like waveform after E-4031 treatment (1 μM). **f** Representative polymorphic ventricular tachycardia TdP-like waveform after E-4031 treatment (1 μM)

**Table 1 Parameters of field potential measurements after E-4031 treatment in CTSs with 50% cardiomyocytes (±s.d., n = 4)**

|  | Beat rate (/min) | ISI (ms) | FPD (ms) | First peak amplitude (μV) | Conduction velocity (mm/s) |
|---|---|---|---|---|---|
| 0 nM | 28 ± 12 | 2591 ± 1244 | 376 ± 172 | 498 ± 211 | 43 ± 24 |
| 0.3 nM | 27 ± 12 | 2656 ± 1265 | 384 ± 179 | 511 ± 211 | 44 ± 24 |
| 3 nM | 26 ± 14 | 2743 ± 1343 | 400 ± 183 | 524 ± 221 | 46 ± 22 |
| 10 nM | 28 ± 14 | 2609 ± 1284 | 422 ± 176 | 537 ± 217 | 46 ± 24 |
| 30 nM | 31 ± 14 | 2346 ± 1094 | 424 ± 152 | 541 ± 213 | 43 ± 26 |
| 100 nM | 29 ± 13 | 2412 ± 1078 | 482 ± 154 | 536 ± 210 | 48 ± 19 |
| 1 μM | N/A | N/A | N/A | 522 ± 218 | 42 ± 19 |
| 2 μM | N/A | N/A | N/A | 540 (n = 2) | 53 (n = 2) |

ISI interspike interval. N/A not available

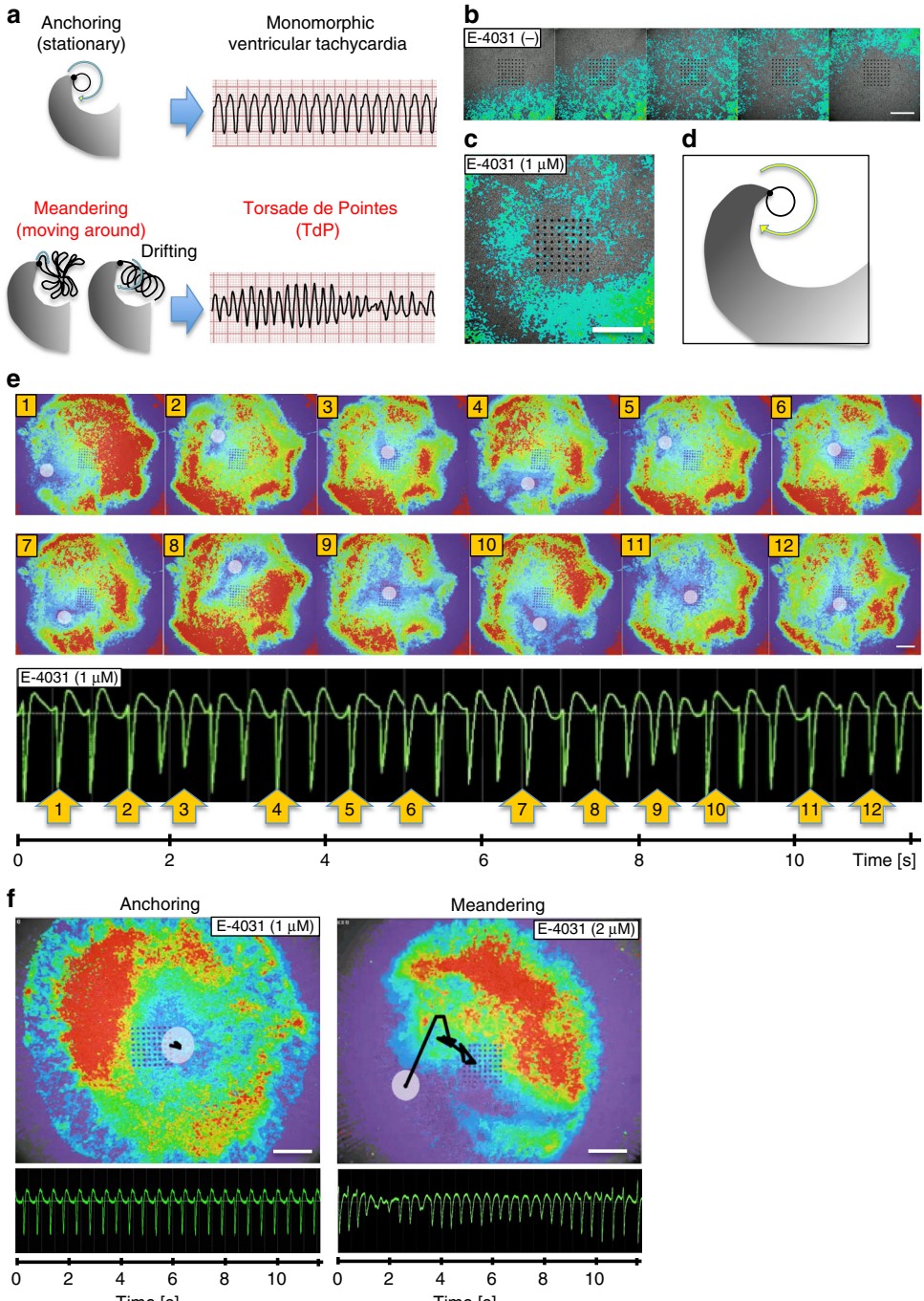

**Fig. 3** Drug-induced TdP in CTSs visualized by Motion Vector Prediction. **a** Schematic images of two re-entrant arrhythmia patterns for the spiral wave re-entry theory. Anchoring pattern and Meandering pattern. **b** Representative normal beats visualized by motion vector prediction (captured from Supplementary Movie 3). Unidirectional biphasic wave propagation (E-4031, 0 nM). **c** Representative anchoring spiral wave re-entry visualized by motion vector prediction (captured from Supplementary Movie 4). Anchoring spiral wave re-entry motion propagation (E-4031, 1 μM). **d** Schematic image related to **c**. **e** Representative meandering spiral wave re-entry with simultaneous recording of EFP and cell motion (E-4031, 1 μM). Cell motion is visualized by motion vector prediction (captured from Supplementary Movie 6). White circles: spiral wave center. EFP shows TdP-like waveform. The numbers in the upper and lower panels indicate the same time points. Note that similar spiral wave center movement (ex: 2 to 3 and 5 to 6) causes similar EFP polymorphisms. **f** Tracing of the spiral wave centers. Representative anchoring of spiral wave re-entry with the corresponding monomorphic ventricular tachycardia-like EFP waveform (E-4031, 1 μM) and representative meandering (drifting) of spiral wave re-entry (upper) with the corresponding TdP-like EFP waveform (E-4031, 2 μM). White circles: center of spiral wave. Black lines: trace of the center. Scale bars: 1 mm

imaging system (Motion Vector Prediction; Sony, Tokyo, Japan) that captures the magnitude of beating cell motion (contraction and relaxation) as biphasic waves (Supplementary Fig. 7) and visualized computed 2D images of the projection map. We simultaneously recorded EFP and cell motion imaging and

successfully detected the appearance of both spiral wave re-entry patterns. Before E-4031 treatment, we confirmed unidirectional biphasic wave propagation with simple contraction-relaxation movements of synchronized beating (Fig. 3b and Supplementary Movie 3). After E-4031 treatment, we observed spiral wave

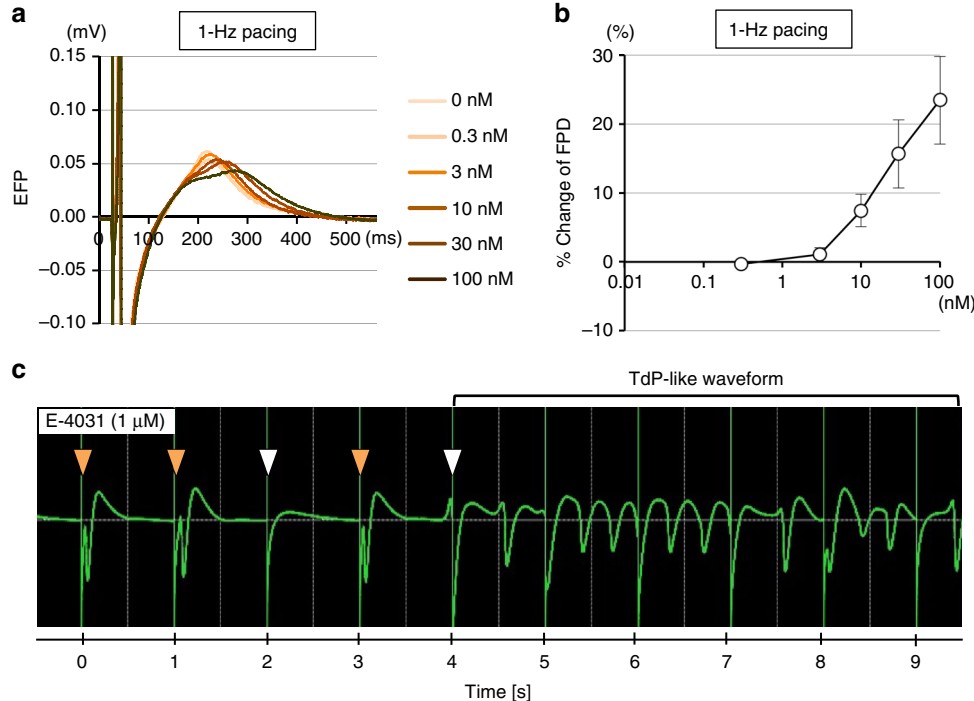

**Fig. 4** Drug-induced TdP-like EFP waveforms in CTSs under 1-Hz pacing. **a** Overlaid EFP waveforms at each E-4031 dosage under 1-Hz pacing. FPD prolongation is observed. **b** Relative change in FPD after E-4031 treatment under 1-Hz pacing ($\pm$s.d., $n = 4$). **c** Representative TdP-like waveforms after E-4031 treatment (1 μM) under 1-Hz pacing. The white arrowhead shows transient pacing failure that is not followed by electrical excitation of the CTS. The orange arrowhead shows pacing success followed by electrical excitation of CTS. The first pacing failure resulted in short-long-short sequence-like waveforms

re-entry waveforms (Fig. 3c, d and Supplementary Movie 4). A transition process from the unidirectional waveform to spiral wave re-entry was also observed (Supplementary Movie 5). Simultaneous recording of EFP and cell motion imaging successfully exhibited polymorphic waveforms of EFP together with 2D spiral cell motion that clearly shows meandering of the spiral wave center (Fig. 3e, f and Supplementary Movie 6–8). These features exemplify the meandering pattern of the spiral wave re-entry that is related to TdP. In contrast, for the anchoring pattern, the center was almost pinned to the 2D motion projection map and was accompanied by monomorphic VT-like EFP (Fig. 3f and Supplementary Movie 8). Voltage-sensitive dye staining of the CTSs also revealed a spiral wave re-entry pattern in the electrical voltage oscillation after E-4031 treatment, indicating that the spiral wave re-entry represented both the kinetic and the electrical behavior of the CTSs (Supplementary Fig. 8 and Supplementary Movie 9). The calculated round frequency (round per second) of the tachyarrhythmias in our CTSs was $2.5 \pm 0.8$ Hz ($n = 4$) in regular sustained tachyarrhythmia and $2.8 \pm 0.3$ Hz ($n = 4$) in TdP-like waveforms. The round frequency was > 8-fold higher than that in a previous report of a re-entrant arrhythmia model using human iPSC-derived cardiomyocytes[10]. Thus, drug-induced TdP-like arrhythmias were successfully reproduced in vitro with our hiPSC-derived CTS model.

**Validation of TdP-like waveform in CTSs**. We confirmed the appearance of TdP-like arrhythmias in another cell line and using other drugs. We used commercially available hiPSC-derived cardiomyocytes, MiraCell® (Takara Bio Inc., Kusatsu, Japan), which were induced from Cellartis human iPS cell line 12 (Cellartis, Göteborg, Sweden), to generate CTSs. These CTSs exhibited similar TdP-like waveforms and meandering spiral wave re-entry in response to as little as 10 nM E-4031 (Supplementary

**Table 2 Change of field potential durations after E-4031 treatment in 1-Hz paced CTSs with 50% cardiomyocytes ($\pm$s.d., $n = 4$)**

|  | Beat rate (/min) | FPD (ms) |
| --- | --- | --- |
| 0 nM | 60 | 199 ± 22 |
| 0.3 nM | 60 | 198 ± 22 |
| 3 nM | 60 | 201 ± 19 |
| 10 nM | 60 | 213 ± 16 |
| 30 nM | 60 | 229 ± 13 |
| 100 nM | 60 | 244 ± 14 |

Fig. 9 and Supplementary Movie 10). We also observed short-long-short sequences in the initiation of TdP-like waveforms, which is a representative pattern of the initiation of TdP in clinical settings (Supplementary Fig. 9b)[30]. We tested cisapride, a gastroprokinetic agent withdrawn from the U.K. and U.S. markets because of reported heart rhythm abnormalities, including sudden death that was possibly related to TdP[31]. Cisapride caused TdP-like waveforms with a high induction rate in CTSs constituting a mixture of cardiomyocytes and non-myocytes (9 of 15; 60%), but no induction was observed in pure cardiomyocyte sheets (Supplementary Fig. 10a, b). Similar TdP-like waveforms were observed after the addition of flecainide, which is clinically reported to induce TdP due to QT prolongation (Supplementary Fig. 10c)[32]. These results indicate that the induction of TdP-like EFP waveforms can be reproduced in CTSs with relevance to the clinical findings.

We also tried to induce TdP-like waveforms under CTS pacing. Under 1-Hz electrical stimulation, the CTSs pulsated in accordance with the pacing frequency. Administration of E-4031 dose-dependently elongated the FPD, as observed in spontaneous beating CTSs (Fig. 4a, b and Table 2). Furthermore,

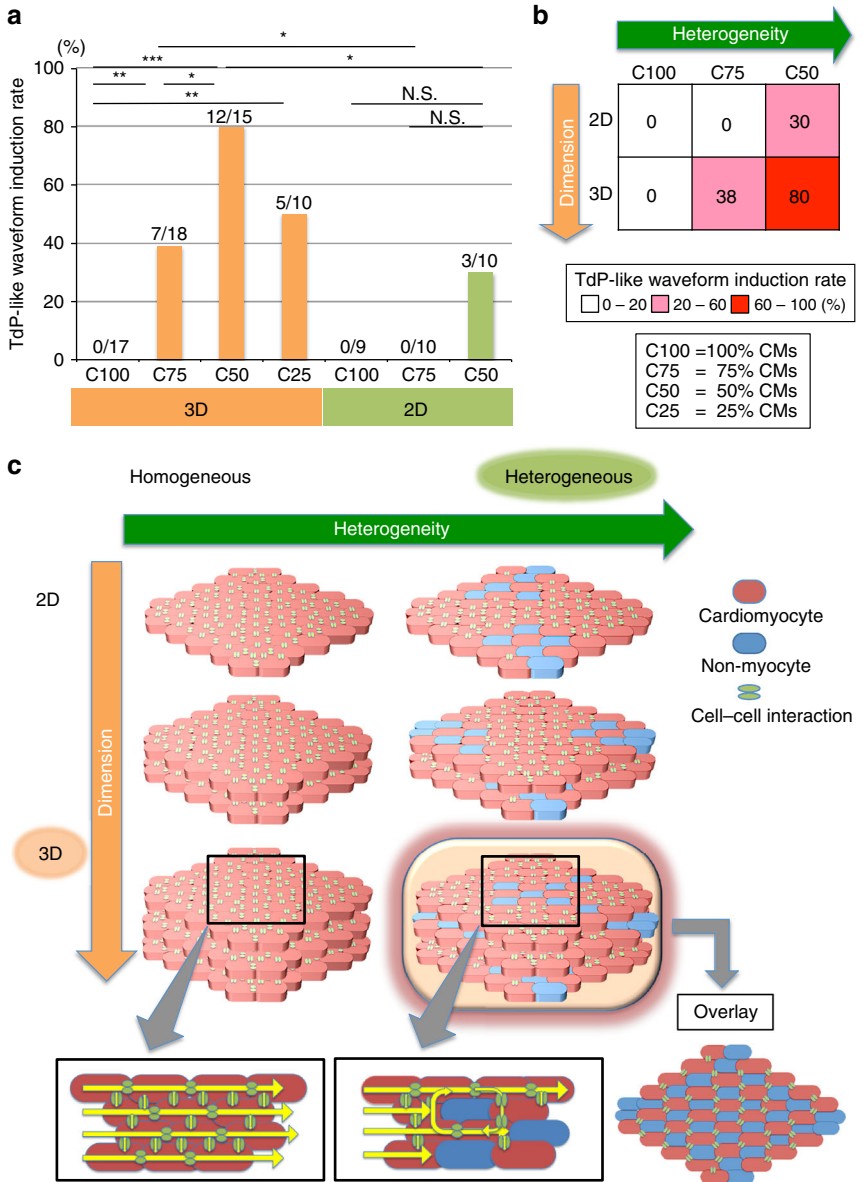

**Fig. 5** Sine qua non for TdP generation in vitro. **a** Induction rates of TdP-like waveform in CTSs of different heterogeneities and dimensions. C100, cells or sheets with 100% cardiomyocytes. C75, cells or sheets with 75% cardiomyocytes. C50, cells or sheets with 50% cardiomyocytes. C25, sheets with 25% cardiomyocytes. CMs, cardiomyocytes. Fisher's exact test *$p < 0.05$, **$p < 0.01$ and ***$p < 0.001$. **b** Relationship of TdP-like waveform incidence with cell heterogeneity and dimension. **c** A putative mechanism for the emergence of TdP. Heterogeneity and dimension enhance the initiation of micro re-entry events and the maintenance of meandering spiral wave centers, respectively, in CTSs. Cellular heterogeneity hampers electrical conduction and detours the propagation direction. The 3D structure eliminates the coarse structures that are present in 2D layers and provides appropriate heterogeneity for a functional center of spiral wave re-entry

we successfully induced TdP-like waveforms in most of the paced CTSs by E-4031 administration (5 of 7; 71%) (Fig. 4c). TdP-like waveforms were triggered by pacing failures that resulted in the formation of short-long-short sequence-like patterns. These results indicate that TdP-like phenomena can be reproduced under comparable action potential durations with pacing in our model.

**Sine qua non for TdP generation in vitro**. Because these results suggest that cellular heterogeneity and 3D structure are important for the reproduction of TdP-like arrhythmias, we next tried to dissect the importance of cellular heterogeneity and 3D structure in the pathogenesis of TdP by manipulating these parameters in

the constructs. We examined four classes of 3D CTSs with different cardiomyocyte composition ratios (25, 50, 75, and 100%) in response to E-4031 treatment (Supplementary Fig. 11a). Whereas neither stationary re-entry waveform nor TdP-like waveform was induced on cell sheets with 100% cardiomyocytes (C100; no heterogeneity), as shown above (Supplementary Fig. 2), all CTSs with cellular heterogeneity (C25, C50, C75) exhibited TdP-like waveforms, with the highest incidence (80%) at 50% cardio-myocytes (C50) (Fig. 5a). In contrast, in 2D culture conditions with 1–2 cell layers (Supplementary Fig. 11b), TdP-like wave-forms were observed in only 30% of 2D culture condition, even with high heterogeneity (C50) (Fig. 5a). Similar results were obtained in CTSs that were constructed with MiraCell® cardio-myocytes (Supplementary Fig. 12a, b). The appearance of TdP-

like waveforms was significantly higher in 3D CTSs compared with 2D conditions ($p = 0.0344$, C50; $p = 0.0302$, C75 Fisher's exact test), indicating that the coincidence of cellular heterogeneity and 3D structure is required and sufficient for the emergence of TdP-like arrhythmias in our in vitro model (Fig. 5a, b and Supplementary Fig. 12a, b). The beat-to-beat variability of the repolarization in sequential recordings of the ECG or EFP is a surrogate risk marker for ventricular arrhythmia[33, 34]. We quantified the variability of FPD in 1 Hz-paced CTSs of different cardiomyocyte composition ratios. The short-term variability (STV) and the coefficient of variation in C50 were significantly higher than those in C100 (Supplementary Fig. 13), further supporting that cell heterogeneity increased the risk of TdP. Connexin 43 (Cx43) immunostaining for gap junctions in the CTSs showed that Cx43 expression was predominantly localized in the cardiomyocyte region within the CTSs (Supplementary Fig. 14). Thus, cellular heterogeneity that hampers electrical conduction and detours the propagation direction may relate the formation of micro-re-entry events that could initiate TdP (Fig. 5c)[35].

**Three-dimensional structure as pathogenesis of TdP.** An apparent difference between the two types of spiral wave re-entry as models of VT-like and TdP-like arrhythmias is the movement of the spiral wave center. Whereas VT-like spiral wave re-entry was observed in a 2D in vitro model in a previous study[10], apparent TdP occurrence was not reported. We speculated that the 3D structure is more effective at reproducing the meandering spiral wave re-entry, making it a more reliable model of TdP. In the previous study[10], VT-like spiral wave re-entry was stably induced after the generation of an acellular region in 2D cultured human iPS cell-derived cardiomyocytes. The wavefronts went around the anatomically fixed obstacle, confirming that meandering will never occur in this model. To achieve meandering of the spiral wave center, functional, rather than anatomically fixed, centers (i.e., excitable but non-excited areas) with appropriate heterogeneity should be broadly generated throughout the CTS. If too large an assembly of cardiomyocytes or non-myocytes ("coarse structure") exists, the spiral wave re-entry would be expected to disappear. To confirm this hypothesis, we conducted Z-stacking of four serial cross-sectional layers of CTSs after Cx43 staining. The coefficient variant of Cx43 expression in each layer was reduced after the Z-stacking (Supplementary Fig. 14b). This result indicates that the 3D structure should be beneficial for minimizing the presence of coarse structures and broadly provide appropriate heterogeneity for a functional center of spiral wave re-entry in CTSs, resulting in sustained TdP-like arrhythmias. Thus, cellular heterogeneity and 3D structure are two sine qua non for TdP arrhythmias because they initiate and maintain the meandering spiral wave center, respectively (Fig. 5c)[19, 35].

## Discussion

In the present study, we generated an in vitro TdP model by efficiently integrating current technological progress in (1) stem cell biology, (2) tissue engineering, and (3) live cell imaging. (1) Our unique hiPSC differentiation method toward cardiovascular cell populations[13, 14] is amenable to collecting highly pure cardiac cellular populations for precise modeling and generating an in vitro model entirely from hiPSCs. (2) We also employed a bioengineered cell sheet technology based on a temperature-responsive culture surface[14], which enabled us to generate 3D tissue structures without special equipment or technological demand. (3) Beyond a 1D linear depiction of FPD (resembling ECG), we could visualize 2D live images of actual cell behavior with a high-precision live cell imaging system. The strong signals

detected by this new cell imaging system allowed us to clearly visualize wavefronts that represent states just after depolarization and wave tails that represent the end of repolarization (Supplementary Movie 11) and to successfully confirm the meandering of the spiral wave center as the cellular kinetics. To our knowledge, the present study is the first to directly visualize the occurrence of TdP-like arrhythmias as actual biological responses in human tissue. 3D heart tissue modeling is expected to change the understanding of TdP. The new evidence here that not only electrophysiological phenomena but also structural features contribute to TdP generation would provide a novel concept in TdP pathogenesis. In vitro 3D tissue modeling with human pluripotent stem cells can be extended to reproduce a variety of tissue or organ responses including patient iPSC research. Considering the heterogeneity of healthy and diseased human hearts regarding cellular components[36], individual cell size and/or maturation level, being able to manipulate various parameters including disease cell components would provide further understanding of cardiac pathophysiology and help explore new therapies and drugs for heart disease

## Methods

**Human iPSC culture and differentiation.** Human iPSCs made with 6-factor episomal plasmid vector (Oct3/4, Sox2, Klf4, L-Myc, LIN28 and Glis1) (line: 836B3)[18] (passage 24–34) and 4-factor (Oct3/4, Sox2, Klf4 and c-Myc) (line: 201B6)[37] (passage 38–48) served as pluripotent cells, which were originally established from skin fibroblasts at Center for iPS Cell Research and Application, Kyoto University, Kyoto, Japan. The 836B3 line was used in all experiments unless stated otherwise.

These human iPSCs were adapted and maintained on thin-coat Matrigel (Growth factor-reduced; 1:60 dilution; BD, Franklin Lakes, USA) in mouse embryonic fibroblast conditioned medium (MEF-CM) supplemented with 4–5 ng/mL human basic fibroblast growth factor (bFGF; WAKO, Osaka, Japan)[13, 38]. Cells were passaged as small clusters once every 4–6 days using CTK solution [0.1% collagenase IV, 0.25% trypsin, 20% knockout serum replacement, and 1 mM $CaCl_2$ in phosphate-buffered saline (PBS)].

Cardiovascular cell differentiation was induced as shown in Supplementary Figs. 1a and 3a. Cells were detached by 3–5 min incubation with Versene (0.48 mM EDTA; Thermo Fisher Scientific, Waltham, USA) and seeded onto Matrigel-coated 6-multiwell plate at 7.0-8.0 x $10^5$ cells/well in MEF-CM with 4-5 ng/mL bFGF for 2–3 days before induction. Cells were covered with Matrigel (1:60 dilution) on the day before induction. To induce mesodermal differentiation, we replaced MEF-CM with RPMI + B27Ins(−) medium (RPMI1640, 2 mM L-glutamine, × 1 B27 supplement without insulin) supplemented with 100 ng/mL activin A (ActA; R&D; McKinley Place NE, USA) for 24 h, followed by 10 ng/mL human bone morphogenetic protein 4 (BMP4; R&D) and 10 ng/mL bFGF.

For cardiomyocyte differentiation, cells were dissociated by incubation with Accumax (Innovative Cell Technologies, San Diego, USA) 5 days after induction, and platelet-derived growth factor receptor-α (PDGFRα)-positive cells were purified with a magnetic activated cell sorter (MACS) (Miltenyi Biotec, Bergisch Gladbach, Germany) using anti-PDGFRα antibody conjugated to allophycocyanin (APC) (R&D) and anti-APC microbeads (Miltenyi) (Supplementary Fig. 1a, b). Purified PDGFRα$^+$ cells were replated onto a Matrigel-coated 6-multiwell plate at 2.5–3.0 × $10^6$ cells/well with 3 mL RPMI1640 medium containing 2 mM L-glutamine, 10% fetal bovine serum (FBS) supplemented with 10 μM XAV939 (Merck KGaA, Darmstadt, Germany), 5 μM IWP4 (STEMGENT, Lexington, USA) and a ROCK (Rho kinase) inhibitor (Y-27632, WAKO; 20 μM). After 1 day in culture, 2 mL of the same medium was added. After 1 more day in culture, the culture medium was replaced with RPMI + B27Ins(+) medium (RPMI1640, 2 mM L-glutamine, ×1 B27 supplement with insulin) supplemented with 0.25 μM XAV939, 0.125 μM IWP4 and 20 μM Y-27632. On day 9 after induction, the culture medium was replaced with RPMI + B27Ins(+) medium. The culture medium was refreshed every other day. Beating cells appeared on days 9–12.

For mesenchymal cell differentiation, cells were dissociated by incubation with Accumax 4 days after induction, and PDGFRα$^+$ cells were purified with a MACS as described above (Supplementary Fig. 3a, b). Purified PDGFRα$^+$ cells were replated onto a Matrigel-coated 6-multiwell plate at 2.0–2.5 × $10^6$ cells/well with 3 mL of RPMI1640 medium containing 2 mM L-glutamine, 10% FBS and 20 μM Y-27632. After 2 days in culture, the culture medium was replaced with RPMI1640 medium containing 2 mM L-glutamine and 10% FBS. The culture medium was refreshed every other day.

**Human iPS cell-derived cardiac tissue sheet formation.** Cardiomyocytes (30–60 days after differentiation) were dissociated by incubation with 0.25%

trypsin (Thermo Fisher Scientific), and mesenchymal cells (15–45 days after differentiation) were dissociated by incubation with Accumax (Innovative Cell Technologies).

Cryopreserved hiPSC-derived cardiomyocytes (MiraCell® cardiomyocytes from ChiPSC12) (Takara Bio Inc., Kusatsu, Japan) were prepared according to the manufacturer's protocol. The cells were thawed in thawing medium (Takara Bio Inc.), plated on 50 μg/mL fibronectin-coated 6-well tissue culture plates (2.0–2.7 × 10$^6$ cells/well), and cultured in maintenance medium (Takara Bio Inc.). After two days of culture, the cells were dissociated by incubation with 0.25% trypsin.

For CTSs consisting of pure cardiomyocytes, the cardiomyocytes were plated onto a 0.1% gelatine-coated 48-multiwell UpCell® at 6.0 × 10$^5$ cells/well with 700 μL attachment medium [AM; alpha minimum essential medium (αMEM; Thermo Fisher Scientific) supplemented with 10% FBS, 5 × 10$^{-5}$ M of 2-mercaptoethanol, 50 units/mL penicillin and 50 μg/mL streptomycin] containing 50 ng/mL VEGF$_{165}$ and 10 μM Y-27632. After 4 days in culture, the cells were moved to room temperature. Within 30 min, cells detached spontaneously and floated in the medium as 3D CTS.

For CTSs consisting of cardiomyocytes and mesenchymal cells, the cells were mixed and plated as above. After 4 days in culture, the cells were moved to room temperature. Within 30 min, cells detached spontaneously and floated in the medium as 3D CTS. CTSs that had holes or macroscopic irregularities were excluded from further evaluation.

**Flow cytometry.** During hiPSC differentiation and CTS generation, cells were dissociated by incubation with Accumax and stained with anti-PDGFRα antibody conjugated with APC (1:40, R&D) or anti-CD90 conjugated with APC (clone 5E10, 1:200, BioLegend, San Diego, USA). To eliminate dead cells, cells were stained with LIVE/DEAD fixable Aqua dead cell staining kit (Thermo Fisher Scientific). Cell surface marker staining was performed in PBS with 5% FBS. For intracellular proteins, staining was performed on cells fixed with 4% paraformaldehyde (PFA) in PBS. Cells were stained with the anti-cardiac isoform of Troponin-T (cTnT, clone 13211, 1:150 Thermo Fisher Scientific) labeled with Alexa-488 using Zenon technology (Thermo Fisher Scientific). The staining was performed in PBS with 5% FBS and 0.75% saponin (Sigma-Aldrich, St. Louis, USA). Stained cells were analyzed on an Aria II flow cytometer (BD, Franklin Lakes, USA). Data were collected from at least 5000 events and analyzed with DIVA software (BD).

**Histological analysis.** For immunofluorescence staining, cardiomyocytes and mesenchymal cells were fixed with 4% PFA. The cells were treated with Protein Block Serum-Free (DAKO, Glostrup, Denmark) and incubated overnight with primary antibodies at 4 °C. Cardiomyocytes were stained for cTnT (anti-cardiac isoform of cTnT, Thermo Fisher Scientific; 1:500), myosin light chain 2V (Proteintech, Rosemont, USA; 1:200) and myosin light chain 2A (Synaptic Systems GmbH, Göttingen, Germany; 1:400). Mesenchymal cells were stained for CD90 (BioLegend; 1:100), vimentin (Abcam, Cambridge, UK; 1:400), alpha-smooth muscle actin (Sigma-Aldrich, 1:500) and calponin (Abcam, 1:250). Anti-mouse Alexa 546 (Thermo Fisher Scientific, 1:500), anti-mouse Alexa 488 (Thermo Fisher Scientific, 1:400), anti-rabbit Alexa 488 (Thermo Fisher Scientific, 1:400), or anti-rabbit Alexa 647 (Thermo Fisher Scientific, 1:200) was used as a secondary antibody. Then, 4′,6-diamidino-2-phenylindole (DAPI; Nacalai Tesque, Kyoto, Japan) was used to stain the cell nuclei. Stained cells were photographed with a confocal microscope (Zeiss LMS 700, Carl Zeiss, Oberkochen, Germany) or all-in-one fluorescence microscopic system (Biorevo BZ-9000, Keyence, Osaka, Japan).

To count MLC2V-positive cells, four random views were selected from MLC2V-, MLC2A- and DAPI-stained sections (×200, original magnification), and the number of MLC2V-positive cells was counted in each view with Biorevo BZ-9000.

CTS immunofluorescence staining was performed similarly. cTnT (Thermo Fisher Scientific; 1:500) and CD90 (BioLegend; 1:100) were used as primary antibodies. Anti-mouse Alexa 546 (1:500) and anti-rabbit Alexa 488 (Thermo Fisher Scientific, 1:400) were used as secondary antibodies. For Cx43 staining, cTnT (Thermo Fisher Scientific; 1:500) and Cx43 (Sigma-Aldrich) (1:200) were used as primary antibodies. Anti-mouse Alexa 546 (1:500) and anti-rabbit Alexa 488 (Thermo Fisher Scientific) (1:400) were used as secondary antibodies. CTSs were stained with anti-human CD90 conjugated with Alexa 647, 1:100 (BioLegend). The stained sheets were photographed with a Nikon A1R MP (Multiphoton + N STORM) (Nikon, Tokyo, Japan) or Zeiss LMS 700 (Carl Zeiss).

To measure the coefficient of variation in Cx43 intensity, we selected four serial cross-sectional layers (5-μm intervals) and an overlaid view (×1000, original magnification) and calculated the coefficient of variation of intensity in line pixels (n = 240) in each layer with ImageJ (U. S. National Institutes of Health, Bethesda, USA). Line pixels with no intensity were excluded from the analysis. Coefficients of variation values were calculated with the following equation:

$$Coefficient\ of\ variation = standard\ deviation(intensity)/mean(intensity) \times 100(\%)$$

**Measurement of EFP and drug treatment.** EFPs were measured with the MED system (Alpha MED Scientific, Osaka, Japan) using a Multi-electrode device (MED) probe with 64 planer 50 μm square microelectrodes arranged in an 8 × 8 grid at 150-μm intervals (MED-P515A).

For 3D CTSs, the probe was sterilized with 70% ethanol and ultraviolet irradiation and coated with 0.1% gelatine (Sigma-Aldrich) before use. A 3D CTS was spread onto the MED probe. The medium was aspirated, and the CTS was incubated at 37 °C. After at least 180 min, medium was added, and the CTS was incubated with alpha minimum essential medium (αMEM) supplemented with 10% FBS, 5 × 10$^{-5}$ M 2-mercaptoethanol, 50 units/mL penicillin, 50 μg/mL streptomycin, 50 ng/mL VEGF$_{165}$, and 10 μM Y-27632 for 2 days. Half of the medium was changed every 2 days. Stable spontaneous EFPs were recorded from 3 days to 21 days after the initial placement.

For 2D culture, the probe was sterilized as above and coated with 50 μg/mL fibronectin (BD) before use. A total of 3 × 10$^4$ cells in 2 μL of medium was spread onto the MED probe and incubated at 37 °C. The rest of the procedure was performed as described above.

We measured EFPs according to a previous report with some modifications[11]. Samples were equilibrated for at least 30 min in a CO$_2$ incubator in 2 mL of fresh medium prior to the measurements. After equilibration, the MED probes were maintained at 37 °C with thermo-control systems and covered with a lid through, which the gas was aerated (O$_2$:CO$_2$:N$_2$ = 20%:5%:75%). EFPs from spontaneously beating samples or 1-Hz paced samples were filtered with a vessel with a 1–1000 Hz bandpass filter using the MED64 System. For stable pacing, samples were stimulated with 2 volt (V) and paced from several microelectrodes. FPD was defined as the interval from the first peak (depolarization) to the second peak (repolarization). After recording the basal state, 2 μL dimethyl sulfoxide (DMSO; Wako) was added, and EFPs were recorded for 10 min. Then, the IKr channel blockers E-4031 (Wako), cisapride (Sigma-Aldrich), or the sodium and IKr channel blocker flecainide (Sigma-Aldrich) were added to obtain the target concentrations, and EFPs were recorded in the same manner as the recordings of DMSO treatment. For E-4031, seven drug concentrations (0.3 nM, 3 nM, 10 nM, 30 nM, 100 nM, 1 μM, and 2 μM) were selected to evaluate dose-dependent effects. For cisapride, seven drug concentrations (3 nM, 10 nM, 30 nM, 100 nM, 300 nM, 1 μM and 2 μM) were selected. For flecainide, five drug concentrations (0.03, 0.1, 0.3, 1, and 50 μM) were selected. At each concentration, the EFP was recorded for 10 min and the FPD values from the last 30 beats were averaged and used as the dataset for FPD and waveform analysis[39].

EFPs were processed with MED64 Mobius software (Alpha MED Scientific) (Fig. 1f). Beat rate, the inter-spike interval (msec), FPD (msec), and the amplitude of the initial sharp positive and negative deflection (first peak; μV) were measured[40].

The electrical propagation map visualization software was produced by Alpha MED Scientific as an application that utilizes Excel 2007 (Microsoft, Redmond, USA). The microelectrode with the earliest negative peak potential was set to time zero, and the negative peak timing of the other microelectrodes was measured. The intervals between the negative peak timing of time zero and each electrode were calculated and virtually visualized as a propagation-mapping image (Fig. 1g). Conduction velocities were calculated from the distance and propagation time between time zero electrode and the maximum time electrode.

To measure STV and the coefficient of variation on EFP, we considered the effect of noise-induced transitions. We selected waveforms at three channels with the lowest noise among the 64 channels under 1 Hz pacing. The FPD values from the 30 beats were averaged in the selected three channels and used to calculate STV and the coefficient of variation. STV and the coefficient of variation values were calculated with the following equations:

$$STV = \sum |FPDn + 1 - -FPDn|/N \times \sqrt{2}(ms) \qquad (1)$$

$$Coefficient\ of\ variation = standard\ deviation(FPD)/mean(FPD) \times 100(\%) \qquad (2)$$

**Definition of Torsade de Pointes (TdP) -like waveform on EFP.** TdP-like waveforms satisfied the following two criteria.

- A continuous and characteristic twisting EFP waveform with variation in polarity.
- Continuous changes in the excitation interval corresponding to twisting changes of the EFP waveform. When meandering of the wave center occurs, the motion of wave center chronologically changes the distance between wave center and each electrode, which leads to the continuous changes in excitation interval.

**Motion vector prediction (MVP) analysis.** We used a high-precision live cell motion imaging system (Motion Vector Prediction; Sony, Tokyo, Japan). We recorded the movies in a resolution of 2048 × 2048 pixels and adjusted the frame rates in the range from 18 to 150 images per second. Motion vectors of beating cells and CTSs were calculated using a block-matching algorithm[41]. From the processes of motion detection and analysis, we obtained the deformation speed as positive values. We measured the chronological fluctuation of the motion vector of 262144 points consisting of 4 × 4 pixels as biphasic waveforms, which indicated contraction and relaxation (Supplementary Fig. 7). We visualized the motion amplitude by colour mapping all points included in the view field and analyzed the two dimensional propagation of cellular motion.

**Simultaneous recording of EFP and MVP**. Simultaneous recordings of the motion and EFP were performed using a stage top mini-incubator (INUSFP-MED-F1-PT, Tokai Hit Co., Ltd., Shizuoka, Japan), which can maintain an MED probe in a humidified atmosphere of 95% air / 5% $CO_2$ at 37.0 °C. This mini-incubator was placed on an inverted microscope (Eclipse Ti, Nikon) with an $x-y$ scanning stage (Bios-T, Sigma Koki, Tokyo, Japan). The MED probe was connected to an amplifier system (MED64 System). The round frequency was calculated from EFPs related to spiral wave re-entry motion propagation in the MVP.

**Membrane potential oscillation analysis**. For membrane potential oscillation analyses, CTSs were loaded with FluoVolt (Thermo Fisher Scientific) for 30 min. FluoVolt fluorescence (excitation at 535 nm and emission at 522 nm) of a beating CTS was measured every 33 msec and visualized using Nikon A1R MP (Nikon). Measurements were performed at 37 °C in an incubator system. E-4031 was administered in the medium, and the membrane potential oscillation was recorded.

**RNA extraction and quantitative reverse transcription polymerase chain reaction (RT-PCR)**. Total RNA was extracted from cell sheets using RLT Lysis buffer (QIAGEN Sciences, Maryland, USA) according to the manufacturer's instructions. We purchased human total heart RNA (Takara Bio Inc.). Reverse transcription was performed with ReverTra Ace reverse transcriptase (TOYOBO Co., Osaka, Japan). Quantitative RT-PCR was performed using TaqMan® Gene Expression Master Mix (Thermo Fisher Scientific) according to the manufacturer's instructions. The amount of target RNA was determined with TaqMan® assays (Thermo Fisher Scientific). RRN18S (18S ribosomal RNA) levels were used to normalize the expression data for each gene of interest. The TaqMan® Assay IDs of the target genes are shown below. Gene Symbol/Assay ID: SCN5A/Hs00165693_m1, CACNA1C/Hs00167681_m1, KCNH2/Hs04234270_g1, KCNQ1/Hs00923522_m1, KCNJ2/Hs01876357_s1, KCNJ12/Hs00266926_s1, TNNT2/Hs00943911_m1, 18S rRNA/Hs99999901_s1, GAPDH/Hs02758991_g1.

**Live/dead assay**. A cell sheet was used to evaluate cell viability. To make cell sheets, we used cardiomyocytes derived from the human iPSC line 836B3 (passage 30) and mesenchymal cells derived from the human iPSC line 201B6 (passage 41). The cell sheets were incubated with staining solution (50 mL/L ethidium homodimer I and 50 mL/L Hoechst 33342, PromoCell GmbH, Heidelberg, Germany) in a pH-adjusted buffer for 30 min at room temperature and protected from light. Fluorescent images were obtained using a microscope (Biorevo BZ-9000, Keyence, Osaka, Japan). For several random areas under × 200 magnification, the numbers of ethidium homodimer I-positive cells and Hoechst 33342-positive cells were calculated with Biorevo BZ-9000 software.

**HIF1α staining**. The HIF1α staining of CTSs and cells was performed similarly to the immunofluorescence staining above. To make cell sheets, we used cardio-myocytes derived from 836B3 (passage 30) and mesenchymal cells derived from 201B6 (passage 41). For normal culture, those cells were mixed and plated onto a 0.1% gelatine-coated 96-multiwell glass-bottom plates at $4.0 \times 10^4$ cells/well. An antibody against HIF1α (GeneTex Irvine, USA 1:200) was used as the primary antibody. Anti-rabbit Alexa 488 (Thermo Fisher Scientific) (1:400) was used as the secondary antibody. The stained sheets and cells were photographed with Zeiss LMS 700 (Carl Zeiss).

**Statistical analysis**. All data analyses were performed using JMP version 11.2.0 (SAS Institute, Cary, USA). Comparisons between two groups were made with Wilcoxon signed-rank test. Comparisons among three or more groups were performed using Steel's test. Values are shown as the mean ± s.d. Fisher's exact test was calculated to compare the induction rate. $p$-values < 0.05 were considered significant.

**Data availability**. All data supporting the findings of this study are either included within the article and its Supplementary Information files or are available upon request from the corresponding author.

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

## Acknowledgements

We thank Dr. M. A. Laflamme (University of Toronto) for the detailed protocol for inducing cardiac differentiation from human embryonic stem cells. We also thank Dr. K. Okita (Kyoto University) for a human iPSC line (836B3), Mr. S. Kihara and Mr. S. Miyake (Kyoto University) for technical assistance, and Dr. Peter Karagiannis (Kyoto University) for critical reading of the manuscript.

## Author contributions

M.K. designed and conducted the experiments, analyzed the data and wrote the paper. H.M. designed and conducted the experiments and wrote the paper. G.M. and H.F. conducted the experiments. R.S. supervised the experiments. T.A. analyzed the EFP and motion vector projection data and supervised the experiments. J.K.Y. designed the experiments, analyzed the data, wrote the paper, and supervised the project.

## Additional information

**Competing interests:** J. K. Yamashita is a founder, equity holder, and scientific adviser of iHeart Japan Corporation. J. K. Yamashita and H. Masumoto are co-inventors on multiple pluripotent stem cell-related patents. This study was funded by research grants from the Japan Agency for Medical Research and Development, iHeart Japan Corporation, Nippon Boehringer Ingelheim Co., Ltd. and Takara Bio Inc. The remaining authors declare no competing financial interests.

