## [Peer Review File · Nature Communications]

Editorial Note: Parts of this peer review file have been redacted as indicated to maintain the confidentiality of unpublished data. When text is deleted in rebuttals and referee reports, add "[redacted]" in that location.

Reviewers' comments:

Reviewer #1 (Remarks to the Author):

The manuscript by Kawatou et al is very interesting describing the creating of TdP arrhythmia with iPSC derived cells. TdP arrhythmia is a special type of ventricular arrhythmia associated with LQTS and drug induced cardiac side effects due to blocking the hERG channel. Thus far, there are many papers describing prolongation of repolarisation time of iPSC-derived CMs, but no TdP type of arrhythmia have been presented.

The current paper described that cellular heterogeneity is required as well as 3D culture for TdP to appear.

the paper is well written, but there are some concerns with the paper:

- I would recommend some further discussion about the potential mechanism of TdP. why 3D is required? Similar spiral wavefronts have been presented with atrial type of cells and those are seen in 2D culture.

- are the cells obtained by the differentiation method mainly ventricular? what percentage? Is 100% ventricular cells requirement for arrhythmia?

- The contraction-relaxation method is not described clearly. In the actual paper motion imaging system by 'Sony is mentioned, but in the methods section the analysis of the motion is described completely differently. Please clarify.

- the cell sheets had 5-6 cell layers. How was the survival of the cells within the tissue analyzed? Could there be other factors besides just mixed population of cells to cause the abnormalities in beating? e.g- different O₂ levels or nutrient levels?

- It is quite confusing that different proportions of CMs and MSCs cells are used in different experiments, e.g. ext fig 4. Why C75 is used for Cisapride and C50 for E4031? also the C75 and C50 should be spelled out in the figure. And they should be spelled out in all figures.

Reviewer #2 (Remarks to the Author):

General comments

In this manuscript, Kawatou and coworkers present a comprehensive study aimed at demonstrating that IKr blocker can induce TdP in heterogeneous sheets of human cardiac and non-cardiac cell. This group of superb scientists demonstrate prominent knowledge in stem cell biology, and excellent technical skills in operating diverse experimental systems, combining state-of-the-art technologies. However, these fine techniques which the authors successfully employ, are the focus of this manuscript, while the physiological/electrophysiological aspects of this work are rather weak, as detailed below. My impression is that the authors were somewhat 'carried away' by their exciting technologies, on the expense of providing a strong physiological basis for their study. Finally, the authors need to show very clearly the novel aspects of their work.

Specific comments

- A working hypothesis is missing/needed
- Page 3, line 38: Should be: treatment with.....
- Page 4, lines 43-47: The sentence is too long, should be shortened.

- Page 5, line 62: explain ICH S7B.
- Page 6, lines 73-74: Explain: ".....transient.....per se.
- Page 6, line 80: Specify your unique technologies.
- Page 7, line 102: Here and elsewhere - the highest concentration tested according to your figures is 100 nM! This discrepancy requires a clear explanation.
- Page 9, line 132: How did you observe the reentry in Fig. 2d?
- Page 9, 2nd paragraph and Figure 2d: The authors must measure all other electrogram characteristic to show that the effect of the drug was specific? Did the authors measure conduction velocity? Was the effect of 2 μ M E-4031 specific? This issue must be fully addressed. Ideally, voltage clamp experiment should be performed that at 2 μ M E-4031 only IKr was blocked.
- Page 10, 1st paragraph: Why did you use the MiraCell?
- Page 11, line 159: It is hard to see the transition process.
- Page 14, line 213-4: Please explain the statement. What do you mean by "line shape"?
- Figure 1e: Can you calculate the conduction velocity? Was it affected by the drugs?
- Figure 2: My impression is that the drug decreased the 'R-R interval'; please measure and comment.
- Page 38, lines 570-572. You mention 7 concentrations, but the figures only show 5 concentrations – please explain.
- Page 45, Figure 2: Was the activation phase affected by the drug? Was conduction velocity affected?

Reviewer #3 (Remarks to the Author):

This is an innovative paper in which the authors model drug-induced Torsade de Pointes (TdP), the prototypic arrhythmias related to drug-induced long-QT, using an in vitro model of iPS-CMs. The authors use 2D and 3D sheets (methods published elsewhere) of cells utilizing a mixture of iPS-CM with varied ratios mesenchymal cells. Their preparations with 2D and 3D iPS-CM did not cause TdP when exposed to HERG blocking drugs E4031 and cisapride. However, the addition of mesenchymal cells resulted in monomorphic and polymorphic when the cells were exposed to E4031 and cisapride. The overall concept of the study is intended to be able to model drug-induced arrhythmia risk in a cellular model toward promoting drug safety. The paper is mostly clear, the methods are innovative and the findings are very intriguing. The study results are important advancement in the field. There are a few questions for the authors particularly related to the ability to reproduce the work on a broader scale:

1. Can you please explain in the paper if you are pacing the 2D and 3D models for the reported EFD and FPD at baseline and after addition of increasing concentrations of E4031 and cisapride? This appears to be done with the spontaneous rate of the tissue preparation, however, the rate of spontaneous beating is not mentioned nor if there is variability. This is an important distinction because human ventricular cardiomyocyte APD shortens as pacing frequency increases. Did you try to pace the sample to control for this variability? Careful review of your methods and figures does not indicate that this has been done.
2. Perhaps the reason why the iPS-CM alone did not have TdP was related to the lack of pacing protocols that are typically applied to create the short-long-short sequences which classically induce torsades. Was this done? Clarify why or why not in your paper.
3. The specificity of the modeling results is also questionable given lack of a full complement of ionic currents, particularly for repolarization and specifically IK1. Others have employed different techniques (see references listed below) to compensate for the lack of IK1 in iPS-CMs, none of which are employed nor discussed in your paper.

Bett GC, Kaplan AD, Lis A, Cimato TR, Tzanakakis ES, Zhou Q, Morales MJ, Rasmusson RL. Electronic

"expression" of the inward rectifier in cardiocytes derived from human-induced pluripotent stem cells. Heart Rhythm 10: 1903–1910, 2013.

Vaidyanathan R, Markandeya YS, Kamp TJ, Makielski JC, January CT, Eckhardt LL. Human induced pluripotent stem cell derived cardiomyocytes (hiPS-CM's) over-expressing Kir2.1: an improved model for investigating complex inherited arrhythmia syndromes. American J Physiol Heart Circ Physiol. 2016

4. Cellular heterogeneity model is very intriguing. Please explain how this compares to a normal human heart. This is an important point with respect to the relevance of your model to mimic drug induced longQT. Others have suggested that the multi-cellular model decreases specificity meaning that these drugs are known to cause arrhythmias in normal hearts and your model perhaps is more applicable to diseased hearts. Thus promoting this model for drug-safety could be dangerous as it could down-play the drug effect for normal hearts.

5. Why is there such a long pause before the initiation of TdP in video #2? While TdP can be pause dependent, this is a seemingly very long pause for which the TdP is initiated by a different mechanism

For Reviewer #1:

Comment #1:

I would recommend some further discussion about the potential mechanism of TdP. why 3D is required? Similar spiral wavefronts have been presented with atrial type of cells and those are seen in 2D culture.

Response:

Thank you for the comment. As for significance of 3D structure, one of the *sine qua non* for TdP, we added a speculation as below.

As the reviewer suggested, reentrant waveforms were previously observed in 2D cultures of pluripotent stem cell-derived CMs (Kadota, *Eur Heart J.* 2013;34:1147-56). In that model, the authors induced reentrant waveforms by generating an anatomical hole (no CMs areas) in the 2D-cultured CMs, which became the center of the spiral wave reentry. In this model, although spiral wave reentry was observed, the position of the center was fixed and will never move around, because the center was artificially and anatomically made as a hole with non-cell area. Thus, TdP will never occur in this model.

Not just to initiate but to maintain TdP, the functional reentry center should be

formed broadly in the CTS. In other words, during meandering of the reentry center in TdP, ubiquitous heterogeneity in the CTS is essential to allow the movement of the reentry center. Considering the principle mentioned above, too large assembly of CM alone or non-CM alone (“coarse structure”) would not be suitable as a reentry center, and reentry center and wave would be disappeared when the moving reentry center reached such “coarse structure”. In 2D structure, “coarse structure” should be often formed. But in 3D structure, such “coarse structure” would be cancelled by stacking the 2D layers into 3D structure. This is our current speculation.

We have added Supplementary Figure 14 and descriptions regarding these points in the revised manuscript (line 212 – 214, page 15 and line 217, page 15–line 237, page 17).

Comment #2:

Are the cells obtained by the differentiation method mainly ventricular? what percentage? Is 100% ventricular cells requirement for arrhythmia?

Response:

Thank you for the comment. We performed immunostaining of a ventricular cardiomyocyte marker, MLC2V, and an atrial cardiomyocyte marker, MLC2A, for

the human iPSC-derived cardiomyocytes we used to prepare CTSs. We found that 97.3±1.3% (n=4) of cardiomyocytes were MLC2V-positive ventricular cardiomyocytes. Even though almost all of the cardiomyocytes were ventricular type, it is not clear whether 100% ventricular cardiomyocytes is required for TdP occurrence, and need to be further investigated. We added Supplementary Fig. 1 and descriptions regarding this point in the revised manuscript (line 84 – 85, page 7).

Comment #3:

The contraction-relaxation method is not described clearly. In the actual paper motion imaging system by Sony is mentioned, but in the methods section the analysis of the motion is described completely differently. Please clarify.

Response:

Thank you for the comment. In the 2D recording, we analyzed the motion of one point in the 2D plane. From this analysis, we could record the chronological fluctuation of the motion vector as a biphasic waveform, which indicates contraction and relaxation. We visualized the amplitude of the motion by color mapping on all points in the view field. We added new Supplementary Fig. 7 and a description in the Methods section of the revised manuscript (line 479 – 483, page 26).

Comment #4:

The cell sheets had 5-6 cell layers. How was the survival of the cells within the tissue analyzed? Could there be other factors besides just missed population of cells to cause the abnormalities in beating? e-g- different O₂ levels or nutrient levels?

Response:

We appreciate the valuable comment. We evaluated the viability of the CTSs (C100 and C50) using ethidium homodimer-I staining for dead cells and found that the live cell ratio was almost 100% (C100: $98.7 \pm 0.6\%$, C50: $99.0 \pm 0.5\%$, n=4 for each) (new Supplementary Fig. 4). We also found that the proportion of cells expressing hypoxia-inducible factor 1-alpha (Hif1 α) was very small in the CTSs (new Supplementary Fig. 5), indicating that hypoxic condition was almost negligible in the CTSs used in the present study. Although we could not investigate the effects of cell death or hypoxia on abnormalities in beating, it would be possible and should be further examined. We have added new Supplementary Figs. 4 and 5, and descriptions regarding this point in the revised manuscript (line 115 - 117, page 9).

Comment #5:

It is quite confusing that different proportions of CMs and MSCs cells are used in different experiments, e.g. ext fig 4. Why C75 is used for Cisapride and C50 for

E4031? also the C75 and C50 should be spelled out in the figure. And they should be spelled out in all figures.

Response:

Thank you for the comments.

TdP was observed in both C75 and C50 with Cisapride treatment.

[UNPUBLISHED DATA REDACTED BY EDITORIAL AS PER AUTHOR

REQUEST] The most distinct and solid result is that the incidence of TdP was

definitely observed in heterogeneous structure (C75 or C50), while there was no

induction of TdP in C100. We modified Supplementary Fig. 10 in the revised

manuscript to show the comparison between C100 and heterogeneous structure and

added descriptions regarding this point (line 179 – 184, page 13). To clarify the

meaning of C75 and C50, we spelled them out in figures themselves.

[UNPUBLISH DATA REDACTED BY EDITORIAL AS PER AUTHOR REQUEST]

For Reviewer #2:

Comment #1:

A working hypothesis is missing/needed.

Response:

We appreciate the comment. We have added the working hypothesis to the main text

(line 67 – 68, page 6).

Comment #2:

Page 3, line 38: Should be: treatment with.....

Response:

We appreciate the careful reading. We changed “treatment of” to “treatment with”

(line 34, page 3).

Comment #3:

Page 4, lines 43-47: The sentence is too long, should be shortened.

Response:

Thank you for the suggestion. Following the suggestion, we shortened the sentence

(line 36,page 3 – line 40, page 4).

Comment #4:

Page 5, line 62: explain ICH S7B.

Response:

This guideline describes a non-clinical testing strategy for assessing the potential of a test substance to delay ventricular repolarization. It includes information concerning non-clinical assays and integrated risk assessments. We added a description regarding this point (line 50 – 51, page 5) and reference for ICH S7B (new reference #7) in the revised manuscript.

Comment #5:

Page 6, lines 73-74: Explain: ".....transient.....per se.

Response:

Thank you for the comment. Our intent was to explain that assays based on 2D culture could only detect restricted abnormal electrical activities such as prolongation of FPD, early after depolarization (EAD) and/or triggered activity (TA). EAD and TA are not sustained, but observed just transiently and disappeared soon, that is, they are transient phenomena. We changed the description in the revised manuscript (line 61 - 62, page 6).

Comment #6:

Page 6, line 80: Specify your unique technologies.

Response:

We appreciate the comment. Following the suggestion, we specified our unique technologies by describing the systematic induction of various cardiovascular cells from hiPSCs and the bioengineered cell sheet technology to generate 3D tissue-like structure (line 69 – 71, page 6).

Comment #7:

Page 7, line 102: Here and elsewhere - the highest concentration tested according to your figures is 100 nM! This discrepancy requires a clear explanation.

Response:

Thank you for the comment. To observe FPD prolongation by E4031, 100 nM was a sufficient dose. At higher doses than 100 nM, EAD-like irregular waveforms were observed. To precisely evaluate and show FPD prolongation, less than 100 nM looked more appropriate. So, we usually tested up to 100 nM and used the results for FPD prolongation. As for TdP induction, higher doses were often required especially in 836B3 hiPSC line-derived CTSs. As representative TdP waves and reentrants

were observed at 1 uM, we used them for the results of TdP. To avoid such confusion, we detailed the actual E-4031 concentrations in every related figure. Though we more likely observed TdP-like waveforms at higher concentration (1-2 uM) of E-4031 in 836B3 line experiments, we mainly observed TdP-like waveforms at 100 nM in MiraCell experiments, suggesting that the sensitivity in TdP induction might be different among cell lines. We added a description regarding these points in the revised manuscript (line 174 – 179, page 13).

Comment #8:

Page 9, line 132: How did you observe the reentry in Fig. 2d?

Response:

We appreciate the comment. We had not simultaneously recorded the cell motion yet when we obtained the Fig. 2d (new Fig. 2e) result. We show this figure as a representative VT-like wave form that we observed in an initial experimental stage.

To more precisely describe the situation, we deleted “reentrant” (line 135, page 10).

Comment #9:

Page 9, 2nd paragraph and Figure 2d: The authors must measure all other electrogram characteristic to show that the effect of the drug was specific?

Response:

We appreciate the suggestion.

We can understand your concern that TdP induction may be due to non-specific effects and/or multi-channel inhibition of E4031. The specificity of E4031 was strictly evaluated by a previous study (Okada, *Sci Adv* 2015;1:e1400142) (new reference #19). In that study, the effects of E4031 on six major ion currents (I_{Na} , I_{Ca} , I_{Kr} , I_{Ks} , I_{K1} , and I_{to}) were examined with human channel gene-expressing CHO cells. As shown in Fig. 2 and Supplemental Table 1 of the revised manuscript, E4031 was highly specific to I_{Kr} , and 1-2 μ M showed almost no non-specific effects on other currents than I_{Kr} .

However, we further confirmed the specificity of E4031 according to the reviewer's suggestion. Even though we could precisely evaluate first peak amplitude and conduction velocity, other parameters of EFP became unstable after premature beats were observed (Supplementary Table 1). There were no significant changes in the first peak amplitude and conduction velocity with 1-2 μ M E4031 treatment. Thus, we consider that no significant non-specific effects of E4031 were observed with 1-2

uM of E4031. We added the descriptions regarding this point in the revised manuscript (line 96-97, page 8; line 131 – 133, page 10).

To show other electrogram characteristic than sustained tachyarrhythmia, we observed ectopic irregular waveforms preceding the occurrence of sustained tachyarrhythmia. We added these waveforms in new Fig. 2d and added a description regarding this point (line 133 – 135, page 10).

Comment #10:

Did the authors measure conduction velocity? Was the effect of 2 uM E-4031 specific? This issue must be fully addressed. Ideally, voltage clamp experiment should be performed that at 2 uM E-4031 only IKr was blocked.

Response:

We appreciate the important comment. As explained in our response to Comment #9, in response to the reviewer's comment, we measured conduction velocity and other electrophysiological parameters (up to 2 uM of E4031) in Supplementary Table 1. Even though we could not precisely evaluate first peak amplitude and conduction velocity, other parameters of EFP became unstable after premature beats were observed. There were no significant changes in the first peak amplitude and

conduction velocity with 1-2 uM E4031 treatment. We added descriptions regarding this point (line 131 – 133, page 10).

As mentioned in response to the comment #9, administration of 1-2 uM of E4031 would still show specific effect on IKr. In addition to that, in experiments using MiraCell (Supplementary Fig. 9), we could induce TdP-like waveforms under lower E-4031 concentrations, ex: 100 nM, suggesting that TdP can be induced with a specific inhibition of IKr with E4031, and non-specific multi-channel inhibition would not be required.

As for voltage clamp experiments, it was too difficult to perform voltage clamp experiments because we suppose that it is quite difficult to adequately analyze single cardiomyocytes among the 3D CTSs which are diffusely mixed with non-myocytes. It is our future work to clearly show the specificity of high-dose E-4031 on the blockade of IKr. We have added descriptions of the above in the revised manuscript (line 174 – 179, page 13).

Comment #11:

Page 10, 1st paragraph: Why did you use the MiraCell?

Response:

Thank you for the comment. We used MiraCell to confirm the results in different human iPS cell line-derived cardiomyocytes other than 836B3 line. MiraCell cardiomyocytes are induced from hiPSCs of a different origin and derivation method than 836B3 (described in lines 175 - 177, page 13). Examining more than 2 hiPSC lines would be important to confirm the robustness of results. This also corresponds to the comment #2 from Editor.

Comment #12:

Page 11, line 159: It is hard to see the transition process.

Response:

We appreciate the comment.

In our study this time, the resolution of the motion vector system may be insufficient to clearly see the detail of transition process you would like to see. That would be examined in the future study with the technical improvement of imaging system. The movie (Supplementary Movie 5) succeeded in sequentially recording from a situation in which unidirectional waveforms were observed to another

situation in which reentrant waveforms appeared. In such meaning, we consider that the movie showed a transition process from the unidirectional waveform to spiral wave reentry.

Comment #13:

Page 14, line 213-4: Please explain the statement. What do you mean by "line shape"?

Response:

Thank you for the comment. "Line shape" was intended to indicate the single dimensional depiction of ECG or EFP just plotting the relationship between time and voltage. We changed the description in the revised manuscript (line 248 - 249, page 18, line 258, page 19).

Comment #14:

Figure 1e: Can you calculate the conduction velocity? Was it affected by the drugs?

Response:

We appreciate the suggestion. As explained in our responses to Comments #9&10, we calculated the conduction velocities of the CTSs (Supplementary Table 1). As the result, we could observe no obvious effects of the drug administration on

conduction velocities. We added Supplementary Table 1 and a description in the revised manuscript (line 131 – 133, page 10).

Comment #15:

Figure 2: My impression is that the drug decreased the 'R-R interval'; please measure and comment.

Response:

Thank you for the comment. We measured ISI (Inter Spike Interval), which represents “R-R interval” in ECG under various drug concentrations (Supplementary Table 1). We confirmed that drug administration did not largely affect ISI ($2,591 \pm 1,244$ ms at 0 nM; $2,412 \pm 1,078$ ms at 100nM). We also changed Fig. 2a to show more representative waveforms. After premature beats were observed, the R-R interval was found to decrease because the beat rates became faster. We added Supplementary Table 1 and a description in the revised manuscript (line 131 – 133, page 10).

Comment #16:

Page 38, lines 570-572. You mention 7 concentrations, but the figures only show 5 concentrations – please explain.

Response:

We appreciate the comment. As the reviewer pointed out, Fig. 2a-c shows only 5 concentrations (0.3 – 3 – 10 – 30 –100 nM), but TdP-like waveforms are mainly observed at higher concentrations (1 or 2 uM), which is why we tested 7 concentrations. We did not show the data of EFP or FPD at higher concentrations because of unstable waveforms at these concentrations. We clarified the concentrations of E-4031 in the figure legends and figures themselves of the revised manuscript (new Fig. 2d-f, Fig. 3b-d, new Supplementary Fig. 2d, Supplementary Fig. 8a, Supplementary Fig. 9a, Supplementary Fig. 10a,c).

Comment #17:

Page 45, Figure 2: Was the activation phase affected by the drug? Was conduction velocity affected?

Response:

Thank you for the comment. We measured the first peak amplitude, which indicates the amplitude of depolarization (activation) phase, and the conduction velocity (Supplementary Table 1). We did not observe significant attenuation of either with drug administration. We added Supplementary Table 1 and a description in the

revised manuscript (line 131 – 133, page 10).

Comment #18:

Finally, the authors need to show very clearly the novel aspects of their work.

Response:

We appreciate the comment. As the Editor also pointed out, it is very important and we showed a potential usefulness of the 3D iPSC-engineered heart tissue by providing a potential new biological insight into the pathogenesis of Torsade de Pointes (line 217, page 15 – line 237, page 17) and evidences that the bioengineered tissue can be employed in clinical applications (line 179, page 13 – line 188, page 14) to clarify the novel aspects of this work. This work, to our knowledge, is the first to visualize the occurrence of TdP-like arrhythmia directly as actual biological responses in human tissue (line 254, page 18 – line 256, page 19) achieved by efficient integration of current technological progress in 1) stem cell biology, 2) tissue engineering, and 3) live cell imaging (line 241 - 254, page 18).

Comment #1:

Can you please explain in the paper if you are pacing the 2D and 3D models for the reported EFD and FPD at baseline and after addition of increasing concentrations of E4031 and cisapride?

This appears to be done with the spontaneous rate of the tissue preparation, however, the rate of spontaneous beating is not mentioned nor if there is variability.

This is an important distinction because human ventricular cardiomyocyte APD shortens as pacing frequency increases.

Did you try to pace the sample to control for this variability? Careful review of your methods and figures does not indicate that this has been done.

Response:

We appreciate the comments. We have been aware that the FPD might be affected by the beating frequency, and pacing the CTSs should make the results more stable and reproducible. Now we are setting up and optimizing the pacing system in our 3D system. Accordingly, we performed the data collection under spontaneous beating in this study. We would like to do experiments with pacing in near future.

Following the reviewer's comments, we measured the actual spontaneous beating rate of the CTSs and found that the beating rate was not affected by the drug administration (Supplementary Table 1). **[UNPUBLISHED DATA REDACTED BY EDITORIAL AS PER AUTHOR REQUEST]** We have added Supplementary Table 1 and a description regarding these data in the revised manuscript (line 131 –

133, page 10).

[UNPUBLISHED DATA REDACTED BY EDITORIAL AS PER AUTHOR REQUEST]

Comment #2:

Perhaps the reason why the iPS-CM alone did not have TdP was related to the lack of pacing protocols that are typically applied to create the short-long-short sequences which classically induce torsades. Was this done? Clarify why or why not in your paper.

Response:

Thank you for the comment. As mentioned in the response for Comment#1, we are still on the way to set up the pacing protocol. We suppose that the creation of short-long-short sequences would be an intriguing option as the reviewer suggests and we will try it in the future investigations.

Comment #3:

The specificity of the modeling results is also questionable given lack of a full complement of ionic currents, particularly for repolarization and specifically IK1. Others have employed different techniques (see references listed below) to compensate for the lack of IK1 in iPS-CMs, none of which are employed nor discussed in your paper.

Response:

Thank you for the comment. Following the reviewer's comment, we performed a qPCR analysis of ion channel gene expression levels including KCNJ2 (Kir2.1) (coding IK1) in our CTS samples and human adult heart (new Supplementary Fig. 6).

We found that the expression levels of most ion channel genes (CACNA1C, KCNH2, KCNQ1 and KCNJ12) in CTSs were almost comparable to those in human heart. However, KCNJ2 and SCN5A in CTSs were lower compared to those in human heart, suggesting the cardiomyocytes in our system are still immature. As advised by the reviewer, overcoming the low expression level of IK1 would be a future direction to polish up our CTS system. We added a description regarding this point (line 117 – 120, page 9), new Supplementary Fig. 6 and the references advised by the reviewer (new references #23 and #24) in the revised manuscript.

Comment #4:

Cellular heterogeneity model is very intriguing. Please explain how this compares to a normal human heart. This is an important point with respect to the relevance of your model to mimic drug induced longQT. Others have suggested that the multi-cellular model decreases specificity meaning that these drugs are known to cause arrhythmias in normal hearts and your model perhaps is more applicable to diseased hearts. Thus promoting this model for drug–safety could be dangerous as it could down-play the drug effect for normal hearts.

Response:

We appreciate the comment. This point is crucial for the application of our model.

As shown in a report about the cellular composition of healthy adult heart (Jugdutt,

Circulation 2003) (Reference Table 2 for Reviewer #3), cardiomyocytes are 25%

of total cells by cell number which may indicate that the majority of normal heart is

composed of non-myocytes in the context of cell number. However, the cellular

volume of cardiomyocytes is 75%, which implies that individual cardiomyocytes

are larger than other cell types and that they are more mature than in our system.

Shown in Fig. 1a and b of the revised manuscript, the cell size of cardiomyocytes

and non-cardiomyocytes (CD90-positive) was not so different in our system,

suggesting that in C75 CTS (Fig. 4a) the cardiomyocyte volume may be near 75%.

Nevertheless, it is still difficult to define what situation is similar to the normal

heart because various factors including total volume of each cellular component,

individual cellular size and number, and the maturation level of individual cells and so on are different and would affect the cardiac physiology. Then, we consider that currently it is not appropriate to determine what is normal or disease situation. Now we are planning to apply this system to disease iPSCs such as cardiomyopathy. In addition, we need to further investigate the balance to generate CTSs and try to examine more reagents in our system. By comparing results between diseased and healthy iPSCs and by comparing our system and clinical outcomes, it would become more clear how much is our system suitable for drug-safety tests. Further studies would be needed. Thank you for critical suggestion. We have added a description regarding this point (line 259 – 261, page 19) and a reference (new reference #34) in the revised manuscript.

Reference Table 2 for Reviewer 3

TABLE 1. Myocytes and Nonmyocytes in the Myocardium

Group	By Cell No.	By Cell Volume	By Cell Mass
Cardiomyocyte	25% ¹⁸	=75% ¹⁸	
	30–35% ¹⁹	=67% ⁶	=90% ^{17,20}
	33% ⁶	67% ²²	
		80% ²³	
Nonmyocyte	75%* ¹⁸	=33% ⁶	=10% ^{17,20,21}
	65–70% ¹⁹	33%† ²²	(90–95% fibroblasts)‡ ^{17,20}
	67% ⁶	20% (13% vascular)§ ²³	

*Connective tissue nuclei.

§Includes lumen (volume fraction).

†Mostly fibroblasts.

‡Fibroblasts as % of nonmyocyte fraction.

(Jugdutt, *Circulation* 2003;108:1395-1403)!

Comment #5:

Why is there such a long pause before the initiation of TdP in video #2? While TdP can be pause dependent, this is a seemingly very long pause for which the TdP is initiated by a different mechanism.

Response:

Thank you for the comment. For the reviewer's reference, we attach a longer version of the EFP recording of video#2 (Reference Figure 3 for Reviewer #3) . We just prepared the video (movie) #2 to show a more representative TdP-like waveform. Because TdP-like waveforms are often observed even in the absence of such pauses, we considered the long pause occurred by chance after a preceding long reentrant

beating.

Reference Figure 3 for Reviewer 3

We have responded to your suggestions as sincerely as possible, and we believe that we were able to address all of your comments satisfactorily. We appreciate your constructive and suggestive comments contributed to markedly improve the quality of our manuscript. We hope you will find the revised manuscript suitable for publication.

Sincerely yours,

Jun K. Yamashita, MD, PhD.

Reviewers' comments:

Reviewer #1 (Remarks to the Author):

The authors have adequately responded all my concerns

Reviewer #2 (Remarks to the Author):

General comments

The authors have addressed most of my comments. Few remaining comments still need to be addressed.

Specific comments:

Comment # 7: The response is not sufficiently clear, and contains too many vague terms which should be replaced by decisive terms. For example: "looked more appropriate", "usually", "often" and "more likely".

Comment # 8: This response is unclear. I don't understand the sentence "We had not....result". What do you mean by "initial experimental stage"?

Comment # 9: Neither Fig. 2 nor the Table show that E4031 was highly specific for IKr. Where in Fig. 2 do you see that 1-2 uM showed almost (?) no non-specific effects on other currents than IKr???

Comment # 18: The authors refereed the reviewer to short parts of sentences that do not illustrate the novelty of their work, or the potential clinical application. This response should be improved.

Reviewer #3 (Remarks to the Author):

The short-comings of the paper that have not been addressed nor discussed include:

1. Repolarization is affected by stimulation frequency. As such, at increased rates, the AP duration typically shortens and the opposite occurs with longer stimulation rates. The experiments performed in this paper ignore this physiologic principle. Instead of pacing the preparations to achieve a similar beat frequency, they compare groups of preparations that have variable spontaneous beating rates. Without pacing the preparations for a comparable AP duration between groups, their conclusions regarding effect of drug on repolarization unconvincing.
2. The cornerstone of arrhythmia induction for TdP due to drug-induced long QT syndrome are short-long-short sequences. This was not applied in this paper also due to their inability to pace their preparations. They have claimed a "robust" TdP induction but with only a single example (figure 2, suppl. Movie 2) of TdP spontaneously occurred after a >5 second pause in beating. Figure 2 is deceiving in that the >5 second pause is not shown and only appreciated in the movie. This arrhythmia induction is a different mechanism of arrhythmia induction than is seen in di-LQTS. This is a problem because the intention of the study is to model di-LQTS.

For Reviewer #2:

We deeply appreciate the reviewer's comments. To prepare a manuscript devoid of unclear or vague terms and to improve its clarity and readability, we requested English language editing from Nature Publishing Group Language Editing, which was recommended by the editor.

Comment # 7:

The response is not sufficiently clear, and contains too many vague terms, which should be replaced by decisive terms. For example: "looked more appropriate", "usually", "often" and "more likely".

Previous comment #7:

Page 7, line 102: Here and elsewhere - the highest concentration tested according to your figures is 100 nM! This discrepancy requires a clear explanation.

Previous response:

Thank you for the comment. To observe FPD prolongation by E4031, 100 nM was a sufficient dose. At higher doses than 100 nM, EAD-like irregular waveforms were observed. To precisely evaluate and show FPD prolongation, less than 100 nM looked more appropriate. So, we usually tested up to 100 nM and used the results for FPD prolongation. As for TdP induction, higher doses were often required especially in 836B3 hiPSC line-derived CTSs. As representative TdP waves and reentrants were observed at 1 μ M, we used them for the results of TdP. To avoid such confusion, we detailed the actual E-4031 concentrations in every related figure. Though we more likely observed TdP-like waveforms at higher concentration (1-2 μ M) of E-4031 in 836B3 line experiments, we mainly observed TdP-like waveforms at 100 nM in MiraCell experiments, suggesting that the sensitivity in TdP induction might be different

among cell lines. We added a description regarding these points in the revised manuscript (line 174 – 179, page 13).

Response:

We deeply apologize the inappropriate and vague terms used in the previous response. According to the comment, we have clarified our response as follows:

To observe FPD prolongation by E4031, 100 nM was a sufficient dose in our experimental setup. At doses above 100 nM, EAD-like irregular waveforms were observed. To precisely evaluate and show FPD prolongation, we concluded that less than 100 nM was appropriate, and we tested up to 100 nM and used the results to analyse FPD prolongation. For TdP-like waveform induction, higher doses were required, especially in the 836B3 hiPSC line-derived CTSs. Because representative TdP-like waveforms and re-entrants were observed in response to 1 μ M, we used these waveforms for the results of TdP-like waveforms. To avoid such confusion, we detailed the actual E-4031 concentrations in every related figure. Although we more frequently observed TdP-like waveforms at higher concentration (1-2 μ M) of E-4031 in the 836B3 line experiments, we more frequently observed TdP-like waveforms in response to less than 100 nM in MiraCell experiments, suggesting

that the TdP induction sensitivity might differ among cell lines.

Comment #8:

Comment # 8: This response is unclear. I don't understand the sentence "We had not.....result". What do you mean by "initial experimental stage"?

Previous comment #8:

Page 9, line 132: How did you observe the reentry in Fig. 2d?

Previous response:

We appreciate the comment. We had not simultaneously recorded the cell motion yet when we obtained the Fig. 2d (new Fig. 2e) result. We show this figure as a representative VT-like wave form that we observed in an initial experimental stage. To more precisely describe the situation, we deleted "reentrant" (line 135, page 10).

Response:

Thank you again for the reviewer's critical reading of the response. We meant that we performed only EFP experiments when we started our study (it is what we call "initial experimental stage"). After we obtained the motion vector machine, we started to add the MVP experiments.

Comment #9:

Neither Fig. 2 nor the Table show that E4031 was highly specific for IKr. Where in Fig. 2 do you see that 1-2 uM showed almost (?) no non-specific effects on other currents than IKr???

Response:

We are very sorry for causing confusion.

In the previous response, we mistakenly stated, “As shown in Fig. 2 and Supplemental Table 1 of the revised manuscript, E4031 was highly specific to IKr, and 1-2 μM showed almost no non-specific effects on other currents than IKr.” Our description should have stated “As shown in Fig. 2 and Supplemental Table 1 of the reference paper”.

In the reference paper (Okada, *Sci Adv* 2015;1:e1400142) (reference #19 and Reference Figure 1 for Reviewer 2), the specificity of E-4031 was strictly evaluated. In that study, the effects of E-4031 on six major ion currents (INa, ICa, IKr, IKs, Ik1, and Ito) were examined with human channel gene-expressing CHO cells. This report shows that E-4031 is highly specific for IKr and that 1-2 μM showed almost no non-specific effects on currents other than IKr. I sincerely apologize for our mistake in this description.

Reference Figure 1 for Reviewer 2

Fig. 2. Dose-inhibition relation of drugs on ion currents. Left column (top to bottom): amiodarone, astemizole, bepridil, cisapride, dofetilide, and D-sotalol. Right column (top to bottom): E-4031, moxifloxacin, quinidine, ranolazine, terfenadine, and verapamil. In each graph, relative activities of ion currents are plotted as a function of drug concentration in logarithmic scale. Black line, I_{Na} ; black dotted line, I_{NaL} ; red line, I_{Ca} ; dark blue line, I_{Kr} ; green line, I_{Ks} ; light blue line, I_{K1} ; orange line, I_{to} ; green circle, $ETPC_{unbound}$.

Supple Table.1 Inhibitory actions of 12 drugs on 6 ion channels

Drug	C_{max} ($\mu\text{g/mL}$)	$ETPC_{unbound}$ (μM)	I_{Na}		I_{Ca}		I_{Kr}		I_{Ks}		I_{K1}		I_{to}	
			IC50	h	IC50	h	IC50	h	IC50	h	IC50	h	IC50	h
amiodarone	2.5	0.0007747	0.9754	-0.7462	4.832	-0.8500	0.7557	-0.8169	-	-	-	-	-	-
astemizol	0.004	0.0002878	1.862	-1.532	0.9878	-2.531	0.02812	-1.745	-	-	-	-	-	-
bepridil	1.27	0.03463	0.6465	-1.155	1.455	-2.317	0.1302	-1.461	6.0312	-1.779	-	-	4.521	-1.853
cisapride	0.06	0.002579	2.072	-0.894	4.278	-1.378	0.01472	-1.326	-	-	-	-	-	-
dofetilidie	0.002	0.0016	124.5	-0.329	184	-0.8969	0.03831	-1.983	-	-	-	-	-	-
d,l-sotalol	4.0	14.68	-	-	-	-	356.4	-1.023	-	-	-	-	-	-
E-4031	0.00759	0.005668	737.3	-0.3337	713.7	-0.4539	0.02627	-1.996	-	-	-	-	-	-
moxifloxacin	8.8	10.95	2721.0	-2.173	469.7	-1.549	135.8	-0.9096	2011.0	-1.173	-	-	2624.0	-0.6837
quinidine	7.0	3.235	24.79	-1.3	7.731	-0.8233	0.6377	-1.012	73.33	-1.336	-	-	1.906	-1.198
ranolazine ^{*2}	2.6	2.31	41.08	-0.9037	118.3	-0.8944	3.927	-0.7095	-	-	-	-	433.6	-1.3090
terfenadine	0.0045	0.009	0.4798	-0.5159	0.8615	-2.091	0.09852	-1.176	-	-	-	-	-	-
verapamil	0.4	0.08797	4.272	-0.8503	0.3331	-1.148	0.2013	-1.040	29.88	-0.933	-	-	3.523	-0.9921

*1 When no inhibitory effect was detected, “-” is shown in the boxes.

*2 For ranolazine, the inhibitory effect on late component of I_{Na} was also included using previously reported parameters (Chevalier et al., 2014): $INaL$ IC50=26.26 μM , h=1.5.

(Okada, *Sci Adv* 2015;1:e1400142)

Another review article also indicates that E-4031 specifically blocks IKr at up to 2 μM in the AT-1 (mouse atrial tumour cells), GPV (guinea pig ventricular myocytes), and HAM (human atrial myocytes) cell lines (Tamargo, *Cardiovasc Res* 2004;62:9-33) (Reference Figure 2 for Reviewer 2).

We added a reference related to this point (new #20).

Reference Figure 2 for Reviewer 2

Pharmacological blockade of voltage-gated potassium channels

Drug	I_{to1}	I_{Kur}	I_{Kr}	HERG	I_{Ks}	Species
E4031	>50 μM	>50 μM	10nM	0.17 μM	>100 μM	AT-1, GPV, HAM

Data are expressed as IC_{50} values (concentrations producing 50% inhibition of the current).

AT-1: mouse atrial tumor cells GPV: guinea pig ventricular myocytes HAM: human atrial myocytes

(Referred from Tamargo, *Cardiovasc Res* 2004;62:9-33)

Comment #18:

The authors refereed the reviewer to short parts of sentences that do not illustrate the novelty of their work, or the potential clinical application. This response should be improved.

Previous comment #18:

Finally, the authors need to show very clearly the novel aspects of their work.

Previous response:

We appreciate the comment. As the Editor also pointed out, it is very important and we showed a potential usefulness of the 3D iPSC-engineered heart tissue by providing a potential new biological insight into the pathogenesis of Torsade de Pointes (line 217, page 15 – line 237, page 17) and evidences that the bioengineered tissue can be employed in clinical applications (line 179,

page 13 – line 188, page 14) to clarify the novel aspects of this work. This work, to our knowledge, is the first to visualize the occurrence of TdP-like arrhythmia directly as actual biological responses in human tissue (line 254, page 18 – line 256, page 19) achieved by efficient integration of current technological progress in 1) stem cell biology, 2) tissue engineering, and 3) live cell imaging (line 241 - 254, page 18).

Response:

We appreciate your important suggestions. According to the comment, we changed the response as follows:

One novel aspect of the present work is that our 3D CTS model can provide biological insights into the pathogenesis of TdP. We examined Connexin 43 (Cx43) expression for gap junctions in the CTSs with immunostaining and found that Cx43 expression was predominantly localized in the cardiomyocyte region within the CTSs, resulting in heterogeneous structures within each cell layer in the CTSs (Supplementary Fig. 15). Because TdP never occurs in pure CM sheets, this heterogeneous distribution gap junction pattern is essential for the initiation of the re-entry in the CTSs derived from mixed cell populations. This is the significance of one of the *sine qua non* for TdP induction, cellular heterogeneity. Regarding the significance of the 3D structure, another *sine qua non*, we speculate as follows..

Re-entrant waveforms were previously observed in 2D cultures of pluripotent stem cell-derived CMs (Kadota, *Eur Heart J.* 2013;34:1147-56). In that model, the authors induced re-entrant waveforms by generating an anatomical hole (no CMs areas) in the 2D-cultured CMs, which became the center of the spiral wave re-entry. Although spiral wave re-entry was observed in this model, the position of the center was fixed and cannot move around because the center was artificially and anatomically made as an acellular area. Thus, TdP will never occur in this model. To both initiate and maintain TdP, the functional re-entry center should be broadly formed in the CTS. That is, during meandering of the re-entry center in TdP, ubiquitous heterogeneity in the CTS is essential to allow the re-entry center to move. Based on the above-mentioned principle, an overly large assembly of only CMs or non-CMs (a “coarse structure”) would be unsuitable as a re-entry center, and the re-entry center and wave would disappear when the moving re-entry center reached the coarse structure. In a 2D structure, coarse structures would be formed frequently. In a 3D structure, such coarse structures are eliminated by stacking the 2D layers into a 3D structure. This hypothesis is also supported by a previous computer

simulation study, in which the 3D structure offers an alternative route for the excitation wave, and small-sized obstacles could have a significant effect on wave propagation and spiral dynamics (Tusscher, *Europace* 2007;9 Suppl6:vi38-45) (reference #35). Compatible with this hypothesis, the coefficient of variance of Cx43 expression in each cell layer within the CTS was reduced after the Z-stacking of 4 serial layers of the CTS (Supplementary Fig. 15b). Thus, we speculate that the 3D structure is beneficial to prevent coarse structure formation and provides broad, appropriate heterogeneity for a functional re-entry center in the CTSs, which would maintain TdP for longer periods. This finding is the significance of the second *sine qua non*, 3D structure. Thus, spiral wave fronts can be induced in 2D culture, but TdP, which includes a meandering center, requires a 3D structure. Thus, we believe that our model can provide new biological insights into the pathogenesis of TdP, which addresses the first request of the editor.

We have added Supplementary Figure 15 and descriptions regarding these points in the revised manuscript (lines 226 – 228, page 17 and line 231, page 17–line 251, page 18).

Another novelty of the present work is that our 3D CTS can be further employed in useful application. We added experiments in which drugs other than E-4031 and cisapride were administered to further validate the usefulness of the CTSs from a clinical perspective. Administration of flecainide, which is clinically reported to induce TdP due to QT prolongation (Nasser, *Rev Cardiovasc Med* 2015), resulted in TdP-like waveforms (Supplementary Fig. 10). These results are relevant to clinical findings. In the field of drug safety testing using single cells, APD (or QT) prolongation is a surrogate marker for the risk of TdP occurrence, and drugs related to APD prolongation have not reached the market. However, drugs that induce APD prolongation do not always lead to TdP induction (FDA Briefing Document, Pharmaceutical Science and Clinical Pharmacology Advisory Committee Meeting 2017), and the U.S. Food and Drug Administration (FDA) is aware of the importance of directly confirming the occurrence of arrhythmia, rather than relying on APD prolongation (Zhang, *J Pharmacol Toxicol Methods* 2016 / Gintant, *Nat Rev Drug Discov* 2016). Our 3D CTSs may serve as a novel tool to observe the kinetics of TdP as a tissue behaviour. We have added Supplementary

Fig. 10 and descriptions regarding these points in the revised manuscript (line 184, page 14 – line 193, page 15).

Finally, we propose another novelty of the CTSs. Our 3D CTSs enable the direct observation of the transition into TdP by eliminating the effect of anatomical heterogeneity observed in the actual heart. Even though further technical advances may be required to clearly observe the transition, this feature of our 3D model may provide important biological clues for the development of anti-arrhythmic drugs.

For Reviewer #3:

Comment #1:

Repolarization is affected by stimulation frequency. As such, at increased rates, the AP duration typically shortens and the opposite occurs with longer stimulation rates. The experiments performed in this paper ignore this physiologic principle. Instead of pacing the preparations to achieve a similar beat frequency, they compare groups of preparations that have variable spontaneous beating rates. Without pacing the preparations for a comparable AP duration between groups, their conclusions regarding effect of drug on repolarization unconvincing.

Response:

We appreciate your critical comments. We recognize the importance of pacing on the CTSs, as the reviewer noted, and tried to optimize the conditions to stably pace the CTSs in additional experiments. We succeeded in stably pacing the CTSs by optimizing the parameters, including the amplitude and duration of the electrical charge, and observed field potential duration (FPD) elongation in response to 100 nM E-4031 (Reference figure 1 for reviewer #3).

Reference Figure 1 for Reviewer 3

We administered E-4031 under pacing and found that FPD elongated in a dose-dependent manner, as observed in spontaneous beating (new Supplementary Figure 11a-c).

We added Supplementary Figure 11 and descriptions regarding these points in the revised manuscript (lines 194 – 197, page 15 and lines 423-425, page 25).

Comment #2:

The cornerstone of arrhythmia induction for TdP due to drug-induced long QT syndrome are short-long-short sequences. This was not applied in this paper also due to their inability to pace their preparations. They have claimed a “robust” TdP induction but with only a single example (figure 2, suppl. Movie 2) of TdP spontaneously occurred after a >5 second pause in beating. Figure 2 is deceiving in

that the >5 second pause is not shown and only appreciated in the movie. This arrhythmia induction is a different mechanism of arrhythmia induction than is seen in di-LQTS. This is a problem because the intention of the study is to model di-LQTS.

Response:

Thank you for your critical comment. As the reviewer indicated, the short-long-short sequence is a very important mechanism for the initiation of TdP.

In our model, typical short-long-short sequences were observed at the initiation of TdP. We show representative 3 short-long-short sequence patterns in spontaneous beating conditions in Reference figure 2 for reviewer #3 and added the most representative case in new Supplementary Figure 9. Thus, our model largely reflects the TdP initiation process due to drug-induced long QT syndrome.

Reference Figure 2 for Reviewer 3

Case 1

Long recording period

In addition, we successfully induced TdP-like waveforms under the pacing condition in most CTSs treated with E-4031 (5 of 7; 71%) (new Supplementary Figure 11d and Reference figure 3 for Reviewer #3). This result further supports that TdP-like phenomena can be induced in our model even under a comparable action potential duration.

Reference Figure 3 for Reviewer 3

We added new Supplementary Figure 9 and descriptions regarding these

points in the revised manuscript (lines 181 – 184, page 14 and lines 194 – 202, page 15). We agree that “robust” TdP induction might be overstated and deleted “robust” from our manuscript. We also added a reference (new #30).

REVIEWERS' COMMENTS:

Reviewer #2 (Remarks to the Author):

I have no further comments.

Reviewer #3 (Remarks to the Author):

The stated intent is to modeling diLQTS and TdP (per the introduction). Overall, the authors achieve creation of 3D CTS, recording and mapping TdP-like electrical activity and detail the importance of percentage of non-myocytes used in the preparation. These are important strides forward and this paper has many exciting elements. In this revision, the authors have responded to prior comments from this reviewer by adding requested experiments, which are included in the supplemental data section.

I do not think that additional experiments need to be performed and only have a few comments and one request to make for the manuscript.

There are important distinctions to make with these experiments compared to their goals of modeling diLQTS and TdP.

As previously pointed out by this reviewer, repolarization is affected by stimulation frequency. As such, at increased rates, the AP duration typically shortens and the opposite occurs with longer stimulation rates. The new data confirms this and not only does pacing create more comparable groups, but changes in the cellular physiology are revealed. Pacing of CTS was performed and, as anticipated, the FPD (pseudo action potential duration) shortens with pacing at 1Hz compared to spontaneous beating, and the variance between groups appears to be much less than the non-paced CTS groups, 199 \pm 22ms vs. 376 \pm 172ms, respectively (See Supplemental figure 11c vs. supplemental table 1, n=4 for both groups of CTS). These are important results and highlight the relevance of comparing groups of similar beating frequency and would be of interest in the main paper. As it stands, in the text the authors indicate that the paced CTS FPD increased with 100nM E4031 and TdP like waveform occurred following failure to capture. However, a analysis of Supp. Figure 11, the FPD increased to only 244ms \pm 14 (n=4), thus with a pacing frequency of 1Hz (every 1000ms) the reason why the CTS fails to capture suggests different physiology rather than an excessive FPD. The authors did not perform mapping of the paced CTS to investigate the mechanism of arrhythmic electrical activity. As shown is supplemental table 1, the spontaneous beat rate is extremely slow with a wide variation 26-31/bpm \pm 14 bpm. Thus, their model is one of bradycardic torsades rather than specifically diLQTS related TdP. At the editor's discretion, I would suggest adding these figures and tables into the main body of the paper (including supplemental table 1 and supp. Figure 11).

It is also revealing to see their iPS-CM derivation pathway results in cells enriched in KCNH2 (>2.5 higher than native) and KCNQ1 (2x higher than native) but under-express SCN5A (3-7 x lower than native) and KCNJ2 (undetectable), Supp. Fig 6. This may be the reason for the very short FPD due to this non-physiologic current imbalance. Again, the lack of a cell that mimics human cardiomyocyte physiology significantly limits the ability of to model diLQTS and thus clinically relevant TdP.

Lastly, a minor issue is that in the intro they indicate that "only hiPSC" are used for Tdp activity induction, when in the results there is a series of experiments titrating the hiPS-CMs and non-myocytes.

For Reviewer #3:

Comment #1:

There are important distinctions to make with these experiments compared to their goals of modeling diLQTS and TdP. As previously pointed out by this reviewer, repolarization is affected by stimulation frequency. As such, at increased rates, the AP duration typically shortens and the opposite occurs with longer stimulation rates. The new data confirms this and not only does pacing create more comparable groups, but changes in the cellular physiology are revealed. Pacing of CTS was performed and, as anticipated, the FPD (pseudo action potential duration) shortens with pacing at 1Hz compared to spontaneous beating, and the variance between groups appears to be much less than the non-paced CTS groups, 199 ± 22 ms vs. 376 ± 172 ms, respectively (See Supplemental figure 11c vs. supplemental table 1, $n=4$ for both groups of CTS). These are important results and highlight the relevance of comparing groups of similar beating frequency and would be of interest in the main paper. As it stands, in the text the authors indicate that the paced CTS FPD increased with 100nM E4031 and TdP like waveform occurred following failure to capture. However, analysis of Supp. Figure 11, the FPD increased to only 244 ± 14 ms ($n=4$), thus with a pacing frequency of 1Hz (every 1000ms) the reason why the CTS fails to capture suggests different physiology rather than an excessive FPD. The authors did not perform mapping of the paced CTS to investigate the mechanism of arrhythmic electrical activity. As shown in supplemental table 1, the spontaneous beat rate is extremely slow with a wide variation 26-31/bpm ± 14 bpm. Thus, their model is one of bradycardic torsades rather than specifically diLQTS related TdP. At the editor's discretion, I would suggest adding these figures and tables into the main body of the paper (including supplemental table 1 and supp. Figure 11).

Response:

We deeply appreciate your important comments and suggestions on our findings. Following your suggestion, we upgraded Supplementary Table 1 to new

Table 1, old Supplementary Fig. 11c to new Table 2, old Supplementary Fig. 11a,b,d
to new Fig. 4.

Comment #2:

It is also revealing to see their iPS-CM derivation pathway results in cells enriched in KCNH2 (>2.5 higher than native) and KCNQ1 (2x higher than native) but under-express SCN5A (3-7 x lower than native) and KCNJ2 (undetectable), Supp. Fig 6. This may be the reason for the very short FPD due to this non-physiologic current imbalance. Again, the lack of a cell that mimics human cardiomyocyte physiology significantly limits the ability of to model diLQTS and thus clinically relevant TdP.

Response:

Thank you for your comment. It is our future work to generate CTSs by hiPS-CMs highly recapitulating human CM physiology including ion channel expression which might be much relevant to clinical TdP.

Comment #3:

Lastly, a minor issue is that in the intro they indicate that “only hiPSC” are used for Tdp activity induction, when in the results there is a series of experiments titrating the hiPS-CMs and non-myocytes.

Response:

We are very sorry for causing confusion. In the present study, we used hiPSC-derived CMs and “hiPSC-derived” non-myocytes. Therefore, we described “only with hiPSC-derived cell populations”.